# Heteroaryl derivatives for hole-transport layers improve thermal stability of perovskite solar cells

Hiroyuki Kanda ⬛ ✉, Santa Mondal, Naoto Eguchi ⬛, Naoyuki Nishimura ⬛, Yoyo Hinuma ⬛, Kohei Yamamoto ⬛, Masaki Yumoto ⬛, Kenichi Tashiro, Hideyuki Takada ⬛, Aiko Narazaki ⬛, Takashi Koida ⬛ & Takurou N. Murakami ⬛

The thermal stability of perovskite solar cells (PSCs) remains a critical challenge for their integration into power grid applications. Here, we report the thermally stable PSCs by employing heteroaryl additives in the Spiro-OMeTAD. These additives effectively control void formation in the hole-transport layer and minimize reactivity with the perovskite layer, significantly improving the thermal stability of PSCs at 85 °C. As a result, PSCs with 3-phenylpyridine and 2-phenylpyridine maintained 101% and 104% of their initial photoconversion efficiency after 2400 hours of 85 °C test, respectively. As an added benefit, photovoltaic performance achieved a photoconversion efficiency of 25%. Also, the outdoor test shows 90% of the initial power point voltage after 1570 hours with maximum-power point tracking, showing remarkable light and cycle stability. We revealed the mechanism of how the additive can improve the thermal stability of PSCs by comparing 36 heteroaryl derivatives and 60 additive combinations.

Perovskite solar cells (PSCs) have undergone remarkable advancements since their initial conception, achieving photovoltaic conversion efficiencies exceeding 26%[1–8]. Metal halide perovskites developed by incorporating low-dimensional perovskite[9–18], dopant modification[19–27], passivation technology[28–39], and self-assembling monolayer[40–44]. Exceptional optoelectronic properties, including long carrier lifetimes[45], flexibility[11], and compatibility with facile solution-based fabrication processes, are additional advantages of the PSCs. These characteristics enable PSCs to be a promising candidate for integration into power grid applications as a next-generation solar cell technology after solving stability issue[46–50].

Thermal instability of the PSCs remains a critical challenge for their commercialization. Additives such as 4-*tert*-butylpyridine (tBP) and lithium bis(trifluoromethanesulfonyl)imide (LiTFSI) have been commonly introduced into 2,2',7,7'-tetrakis(*N,N*-di-p-methoxyphenylamine)−9,9'-spirobifluorene (Spiro-OMeTAD) of the hole-transport layer (HTL), where tBP plays an essential role in improving the film morphology and conductivity due to the coordination of tBP and lithium in Spiro-OMeTAD. This improvement from additives

allows for increased HTL thickness (>200 nm), which is important for scalable fabrication techniques such as blade coating for roll-to-roll processing[51,52]. However, under high-temperature conditions, tBP leads to the formation of voids in the HTL due to notorious volatilization, and also causes the reaction with the perovskite layer and compromises the stability of PSCs[53,54]. To improve the stability, several additives and methods were incorporated into the Spiro-OMeTAD layer, such as lithium alternatives[55–64], oxide dopants[65–68], additive engineering[69–82], different doping system[83–86], and stabilization methods[87,88]. Only a few additives were introduced as alternative additives for tBP[89–92], limiting improvement in the stability of PSCs. To date, suitable alternative additives for tBP that ensure thermal durability have yet to be developed.

Heteroaryl derivatives are chemical compounds derived from heteroaryl groups, which are a subclass of aromatic compounds[93–95]. These compounds consist of an aromatic ring structure that includes at least one heteroatom, such as nitrogen, sulfur, and oxygen, within the ring. Derivatives are formed by substituting hydrogen atoms on the heteroaryl ring with other chemical groups, such as alkyl, hydroxyl,

National Institute of Advanced Industrial Science and Technology (AIST), Tsukuba, Ibaraki, Japan. ✉e-mail: hiroyuki.kanda@aist.go.jp

halogen, or functional groups like a phenyl ring. The presence of heteroatoms and these substitutions can significantly alter the chemical properties, including volatilization and reactivity with the perovskite layer. Therefore, by tailoring the chemical structure, heteroaryl derivatives can potentially be designed as alternative additives for tBP to enhance the thermal durability of PSCs.

In this study, we identified heteroaryl derivatives, namely−4-phenylpyridine, 3-phenylpyridine, and 2-phenylpyridine−, as key contributors to the thermal stability of PSCs. The impact of additives on thermal stability at 85 °C was systematically evaluated by comparing 36 heteroaryl derivatives and 60 additive combinations. Additionally, the heteroaryl derivatives enhanced charge extraction from the perovskite layer to the hole-transport layer (HTL), resulting in improved photovoltaic performance of the PSCs. This work provides a promising pathway for developing highly thermally stable PSCs for practical applications.

## Results and Discussion
### Thermal stability results with heteroaryl derivatives
The heteroaryl derivatives examined in this study are presented in Fig. 1 and Supplementary Table 1. These derivatives were categorized based on their structural features: pyridine derivatives substituted at the para position (Fig. 1a), pyridine derivatives substituted at the meta and/or ortho positions (Fig. 1b), para-substituted pyridine derivatives containing a phenyl or hetero ring (Fig. 1c), derivatives featuring phenyl groups substituted at the para, meta, or ortho positions (Fig. 1d) and their substitution (Fig. 1e), and other heteroaryl derivatives, including thiophene, imidazole, and unique structures such as

bipyridine (Fig. 1f). For identification purposes, these additives were assigned code names ranging from K1 to K36. The materials were selected and screened to investigate the overall effect of para and ortho/meta substitutions on thermal durability. Phenyl and heterocyclic rings were included to explore their influence on volatilization and reactivity with the perovskite layer, while thiophene and imidazole derivatives were chosen to evaluate the impact of different heteroatoms on the thermal stability of PSCs. These additives were incorporated into Spiro-OMeTAD along with LiTFSI and/or tris(2-(1H-pyrazol-1-yl)-4-tert-butylpyridine)cobalt(III) tri[bis(trifluoromethane)sulfonimide] (Co(III)TFSI, FK209). Since there are few examples of the additives other than tBP, screening of such materials will be useful for comprehensively understanding the effect of additives on thermal durability. It is desirable to examine the selection of heteroaryl derivatives and their impact on durability from the following perspectives. For the volatility, some literature insists on the volatility of the tBP, which can be the reason for the weak durability of the Spiro-OMeTAD[96]. Thus, it is important to evaluate heteroaryl derivatives with different boiling points to investigate the effects of volatility on thermal stability. For example, the boiling points of 2-phenylpyridine, 3-phenylpyridine, and 4-phenylpyridine correspond to 270 °C, 274 °C, and 281 °C, respectively, which is higher than tBP (197 °C). Also, derivatives with lower boiling points should be selected for comparison, e.g., 4-ethylpyridine (165 °C), 4-aminopyridine (157 - 162 °C). Regarding steric hindrance, Yue et al. introduced 2-amylpyridine instead of tBP, which is substituted with an amyl group at ortho position[91]. They explained that the steric hindrance from the ortho-substitution of the pyridine ring may prevent the formation of a complex with additives

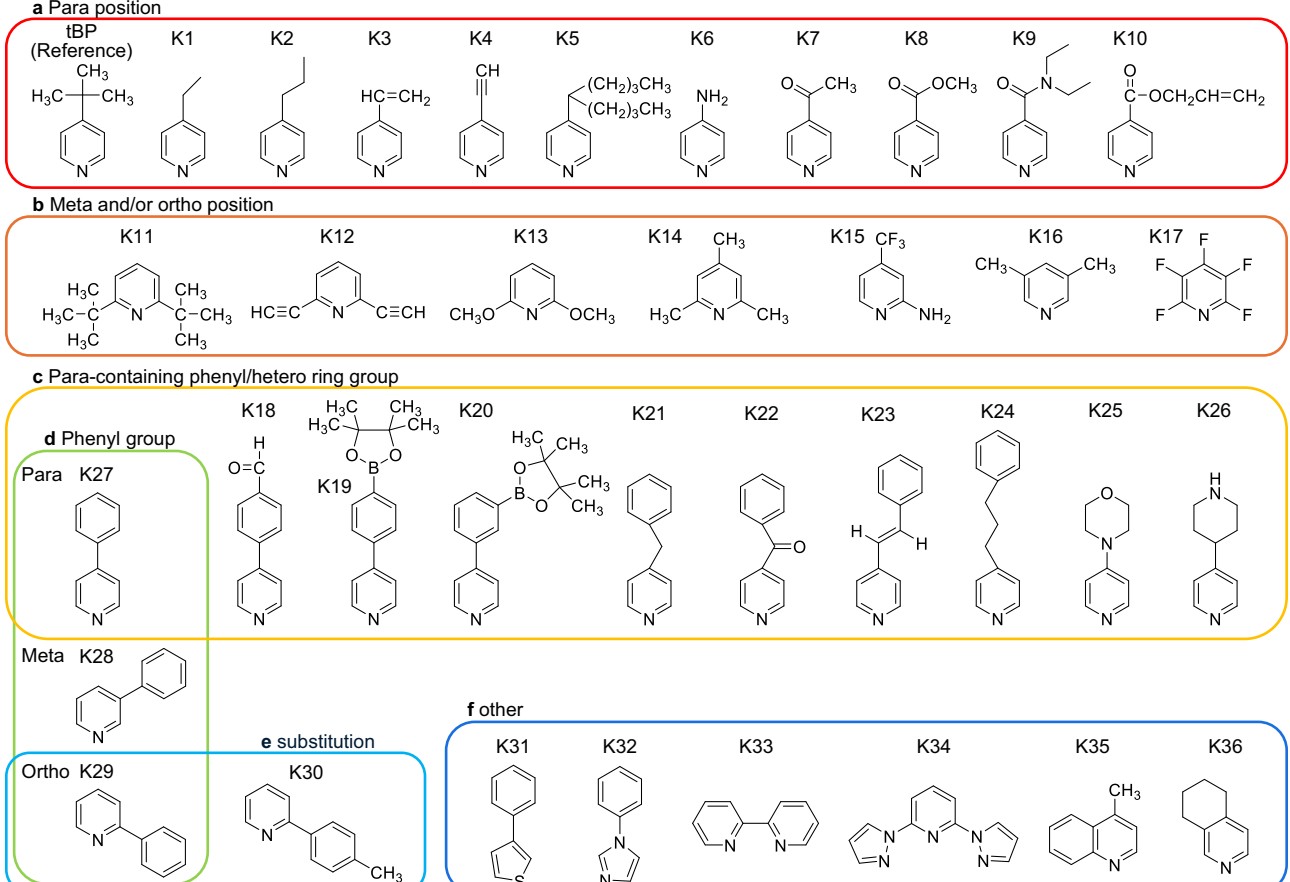

**Fig. 1 | Code and chemical structure of heteroaryl additives for Spiro-OMeTAD.** **a** Pyridine derivatives substituted with a para position. **b** Pyridine derivatives substituted with meta and/or ortho position. **c** Pyridine derivatives with para-containing phenyl/hetero ring group. **d** Phenyl group of para, meta, ortho position. **e** Phenyl group with substitution. **f** Other additives, including thiophene, imidazole, and a unique structure such as bipyridyl.

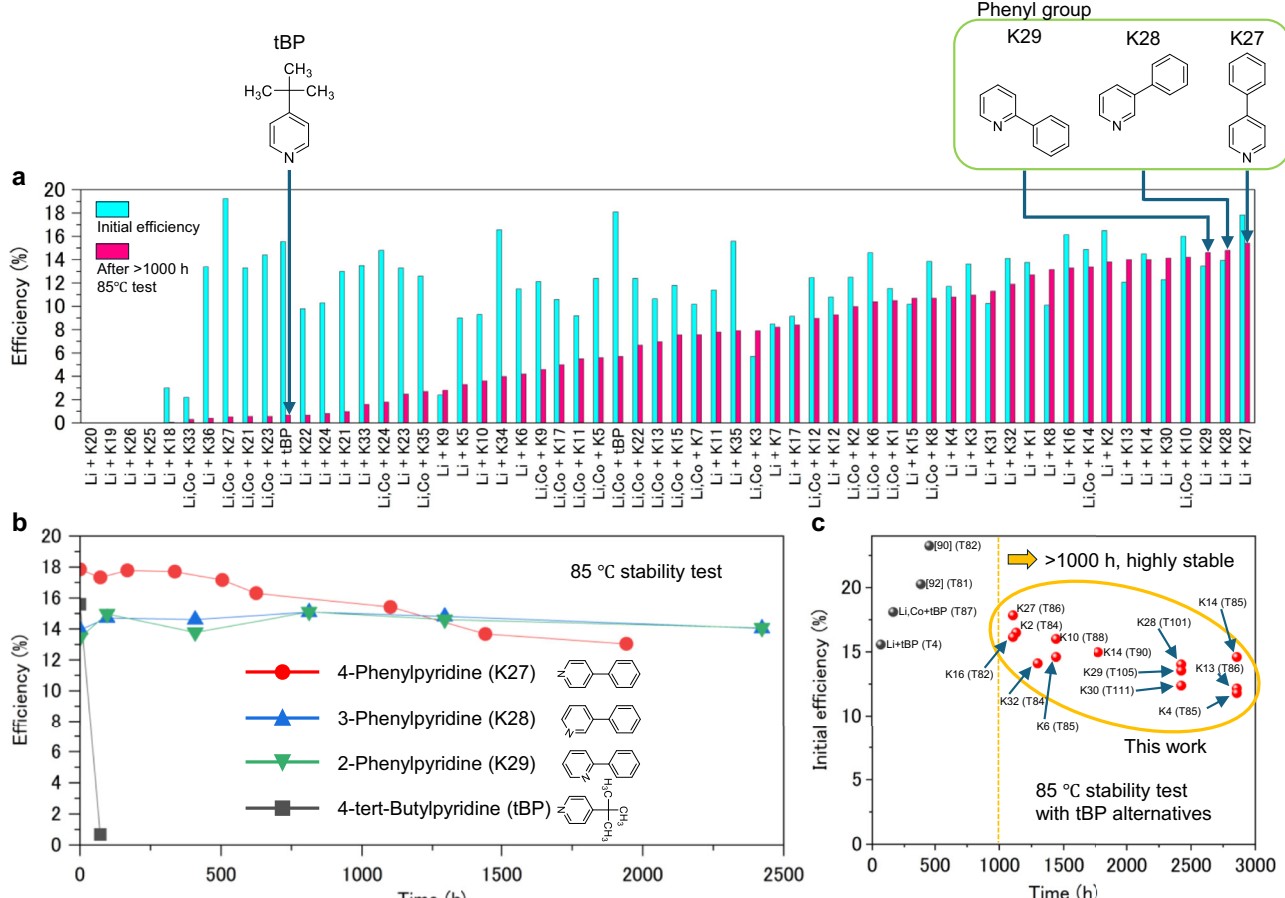

**Fig. 2 | Summarized 85 °C thermal stability results of perovskite solar cells with various additives in Spiro-OMeTAD. a** Stability result compared to initial efficiency and after 85 °C test over 1000 hours as a function of additives. Note that if the stability test was finished before 1000 hours, the last efficiency was plotted according to supplemental information. LiTFSI and Co(III)TFSI are abbreviated as Li and Co, respectively. **b** 85 °C thermal stability result with 4-phenylpyridine (Li +K27), 3-phenylpyridine (Li+K28), 2-phenylpyridine (Li+K29), and 4-*tert*-butylpyridine (Li+tBP). **c** Stability chart.

and perovskite, which could increase the durability of perovskite solar cells. Therefore, we can select ortho, meta, and para-substituted heteroaryl derivatives to investigate the effect of the steric hindrance on the durability. As for the electronic properties, they can be tuned by introducing various functional groups and heteroatoms onto the hetero ring. These functional groups exhibit electron-donating or electron-withdrawing effects, which influence the electronegativity of the pyridine ring and its interaction with lithium. Evaluating these effects is important to determine whether the electronic state influences thermal stability. Therefore, a variety of functional groups with different electron-donating or -withdrawing characteristics, as well as heteroatoms, need to be verified.

The thermal stability results are presented in Fig. 2, Supplementary Figs. 1-67, and Tables 2-61. In these figures and tables, LiTFSI and Co(III)TFSI are abbreviated as Li and Co, respectively. Heteroaryl derivatives were incorporated into Spiro-OMeTAD with LiTFSI (e.g., Li +K1) or co-doped with both LiTFSI and Co(III)TFSI (e.g., Li, Co+K1). Importantly, *n*-octylammonium iodide (OAI) passivation was not used in this study to isolate the intrinsic effects of the additives on thermal stability, avoiding any confounding influences from passivation[97]. While OAI passivation increases the photoconversion efficiency (*PCE*) to 21.6% and decreases the hysteresis (Li+K29: Supplementary Figs. 68, 69), the unpassivated devices—despite their lower efficiency—offer clearer insights into how additives directly impact thermal stability of PSCs. This approach can be vital for effective materials screening and evaluation. Figure 2a shows stability results compared to initial efficiency and after 85 °C thermal stability tests after 1000 hours. Note

that if the stability test was finished before 1000 hours, the final efficiency was recorded as described in the supplemental information.

Regarding the pyridine derivatives substituted at the para-position (Fig. 1a), the categorized summary is presented in Supplementary Fig. 63. For the reference device (Li+tBP; Supplementary Figs. 1, 58, 63, Table 2), the initial *PCE* was 15.6%, but it exhibited a sharp decline to 0.7% after 72 hours of thermal stability testing at 85 °C. This degradation is attributed primarily to reductions in current density (*J*$_{SC}$) from 24.78 mA/cm² to 2.96 mA/cm² and fill factor (*FF*) from 0.626 to 0.238, while the open-circuit voltage (*V*$_{OC}$) remained relatively unchanged. In the case of devices co-doped with Co(III)TFSI, LiTFSI, and tBP (Co, Li+tBP; Supplementary Fig. 2, 59, 63, Table 3) exhibited slightly improved stability, but the overall trend remains similar. This instability is linked to the presence of tBP, which promotes void formation in the Spiro-OMeTAD layer, leading to substantial declines in *FF* and *PCE*[96]. Interestingly, certain pyridine derivatives demonstrated enhanced thermal stability, including allyl isonicotinate (Li, Co+K10), 4-propylpyridine (Li+K2), methyl isonicotinate (Li+K8), and 4-ethylpyridine (Li+K1) (Supplementary Fig. 63). Among these, the device doped with allyl isonicotinate (Li, Co+K10, Supplementary Fig. 21, Supplementary Table 22) retained 89% of its initial *PCE* after 1104 hours in the thermal stability test. Several factors may contribute to these improvements. Regarding additive volatility, the vapor pressure of 4-ethylpyridine (K1) is expected to be lower than that of tBP due to the size of its hydrocarbon group and Antoine constants, indicating that the thermal stability of PSCs with tBP is higher than 4-ethylpyridine (K1)[98]. However, the thermal stability of PSCs with tBP

was significantly lower than that of 4-ethylpyridine (Li+K1), suggesting that additional factors influence the thermal stability of these devices. As for the electronic effects of substituents, substituents of 4-propylpyridine (K2) and 4-*tert*-butylpyridine (tBP) exhibit weak electron-donating properties, while 4-aminopyridine (K6) and 4-acetylpyridine (K7) display strong electron-donating and electron-withdrawing effects, respectively[99]. While methyl isonicotinate (K8) is a weak electron-withdrawing. These electronic properties may locally alter the electron density within the pyridine ring. However, no clear trend correlating these effects with the thermal stability of PSCs was observed.

In terms of pyridine substituted at the meta and/or ortho positions (Fig. 1b, Supplementary Fig. 64), devices incorporating 2,4,6-trimethylpyridine (Li+K14), 2,6-dimethoxypyridine (Li+K13), and 3,5-lutidine (Li+K16) demonstrated improved thermal stability compared to the device with tBP (Supplementary Figs. 26, 28, 32, Supplementary Tables 27, 29, 33). Notably, dopants with simple chemical structures, such as 2,4,6-trimethylpyridine (K14), exhibited greater durability than tBP. The overall trend observed for substituents at the ortho, meta, and para positions indicated minimal changes in $J_{SC}$ and $V_{OC}$ across devices during the thermal stability test, except in cases with poor stability. The *FF* appeared to be the primary parameter associated with the thermal stability of PSCs in these systems.

As for para-substituted pyridine derivatives containing a phenyl or hetero ring (Fig. 1c, Supplementary Fig. 65), devices incorporating 4-phenylpyridine (Li+K27) demonstrated significantly enhanced thermal stability (Fig. 2b, Supplementary Figs. 45, 60, Supplementary Table 49). *PCE* retained 87% of the initial value (*PCE*$_{Initial}$ = 17.8%) after 1,104 hours of thermal test at 85 °C. In contrast, the thermal stability of devices with other additives was markedly low (Supplementary Fig. 65). Interestingly, despite 4-benzylpyridine (Li+K21) having only one additional alkyl chain linking the pyridine and phenyl rings compared to 4-phenylpyridine (Li+K27), its thermal stability was significantly reduced (Supplementary Fig. 36, Supplementary Table 39). This suggests that the steric hindrance of the additive may play a critical role in thermal durability. Moreover, in the case of the co-doping with LiTFSI, Co(III)TFSI, and 4-phenylpyridine (Li, Co+K27), thermal stability was significantly decreased. This can also be related to the steric hindrance influenced by the cobalt complex which is much larger than lithium. These findings highlight the importance of interaction between additives and counterparts of TFSI in the HTL.

Regarding phenyl-substituted pyridines at the para, meta, or ortho positions (Fig. 1d) and their substitution (Fig. 1e), remarkable improvements in thermal stability were observed (Fig. 2a,b, Supplementary Fig. 66). Notably, the *PCE* of devices containing 3-phenylpyridine (Li+K28) and 2-phenylpyridine (Li+K29) were 101% and 104% of their initial *PCE* after 2,424 hours of 85 °C test, respectively (Supplementary Fig. 47, 48, 62, 63, 66, Supplementary Tables 51, 52), demonstrating a slight improvement in the *PCE*. Both $V_{OC}$ and *FF* remained virtually unchanged during the 85 °C test. For 2-(*p*-tolyl) pyridine (Li+K30), the stability was comparable to that of 2-phenylpyridine (Li+K29), indicating that small substituents like a methyl group do not adversely affect the thermal stability of PSCs (Supplementary Fig. 49, Supplementary Table 53). Thermal stability further improved when the phenyl group was substituted from the para to the meta or ortho position (Supplementary Fig. 70). However, 4-Phenylpyridine showed higher *PCE* than 2-phenylpyridine and 3-phenylpyridine for approximately 1,100 hours from the start of the 85 °C stability test. This suggests that 4-phenylpyridine has an advantage in initial *PCE*, which may contribute to greater overall energy output during the degradation period. For example, if 1000 hours accelerated test at 85 °C is sufficient to ensure practical outdoor durability, 4-phenylpyridine could be more advantageous in practical use compared to 2-phenylpyridine and 3-phenylpyridine. Therefore, it is premature to conclude which molecular orientation is the most

superior. We also provided stability comparison data of dispersed-solution-SnO$_2$ (SnO$_2$_DS) vs chemical bath deposition (SnO$_2$_CBD) as well as CsFAPbI$_3$ vs RbCsFAPbI$_3$ (Supplementary Fig. 71). To provide high-efficiency PSCs, we optimized the recipe for perovskite fabrication for each condition, which is explained at the experimental part. In terms of dispersed-solution SnO$_2$ (SnO$_2$_DS) vs. CBD SnO$_2$ (SnO$_2$_CBD), the initial efficiency of SnO$_2$_CBD with 2-phenylpyridine was 20.1% and remained at 18.7% after 936 hours of testing at 85 °C. In contrast, SnO$_2$_DS with 2-phenylpyridine showed an initial efficiency of 14.9% and retained 16.2% after 936 hours at 85 °C. In both cases, 2-phenylpyridine provided higher stability compared with 4-tert-butylpyridine. As for CsFAPbI$_3$ vs RbCsFAPbI$_3$, rubidium-doped perovskite (RbCsFAPbI$_3$) achieved higher efficiency (24.2%) than CsFAPbI$_3$ (20.1%), however, stability with RbCsFAPbI$_3$ showed a significant drop after 87 hours of 85 °C test. From this result, It is possible that rubidium-doped perovskite itself has problems with thermal durability. Correa-Baena et al. indicated that a dosage of 1% Rb was enough for causing segregation, which is a different kind of issue from those associated with additives[100]. These results suggest that optimization of perovskite composition is also important to achieve high thermal stability. A detailed analysis of this phenomenon is provided in the characterization section.

For other heteroaryl additives (Fig. 1f, Supplementary Fig. 67), 1-phenylimidazole (Li+K32) and 3-phenylthiophene (Li+K31) exhibited superior thermal stability compared to 4-*tert*-butylpyridine (Li+tBP) (Supplementary Figs. 50, 51, Supplementary Tables 54, 55). 1-Phenylimidazole (K32) and 3-phenylthiophene (K31) can interact with lithium via the lone pairs on the nitrogen at the 3-position and sulfur at the 1-position, respectively. Given sulfur's lower electronegativity compared to nitrogen, its interaction with lithium is expected to be weaker. Despite these differences in interaction strength, no notable differences in photovoltaic performance or thermal stability were observed between the two additives.

Overall, these results suggest that various heteroaryl derivatives can function as effective additives for Spiro-OMeTAD, enhancing thermal stability relative to tBP (Fig. 2c). Notably, phenyl groups exhibited exceptional improvements in thermal stability, highlighting the need for further investigation into their specific role, as detailed in the following section.

## Mechanism for thermal durability and material characteristics
To examine the impact of additives on void formation in the HTL, SEM measurements were conducted on both as-fabricated devices and those after the 85 °C thermal stability test (Fig. 3). Cross-sectional SEM images of the as-fabricated devices with 4-*tert*-butylpyridine (Li+tBP), 4-phenylpyridine (Li+K27), and 2-phenylpyridine (Li+K29) are presented in Fig. 3a–c, respectively. In the case of the device containing 4-*tert*-butylpyridine (Li+tBP), small voids were observed within the Spiro-OMeTAD layer (Fig. 3a). These voids are likely attributed to the post-annealing process temperature (85 °C, 10 min) following HTL deposition, as described in the experimental section. Conversely, the devices containing 4-phenylpyridine (Li+K27) and 2-phenylpyridine (Li+K29) exhibited void-free HTLs, as shown in Fig. 3b and 3c. Cross-sectional SEM images of the devices after 400 hours of thermal stability test at 85 °C are shown in Fig. 3d-f. For the device with 4-*tert*-butylpyridine (Li+tBP), significantly larger voids were observed within the Spiro-OMeTAD layer (Fig. 3d, Li, Co+tBP, Supplementary Fig. 72). These voids appeared to expand during thermal testing, extending to the HTL/perovskite and HTL/Au interfaces. This extensive structural transformation likely explains the observed decrease in the *FF* during the thermal stability test, leading to the poor thermal stability of PSCs. In contrast, the device containing 4-phenylpyridine (Li+K27) exhibited reduced void formation, though several small voids persisted within the Spiro-OMeTAD layer (Fig. 3e). Additionally, a thin layer of by-products may have formed at the interface between the Spiro-

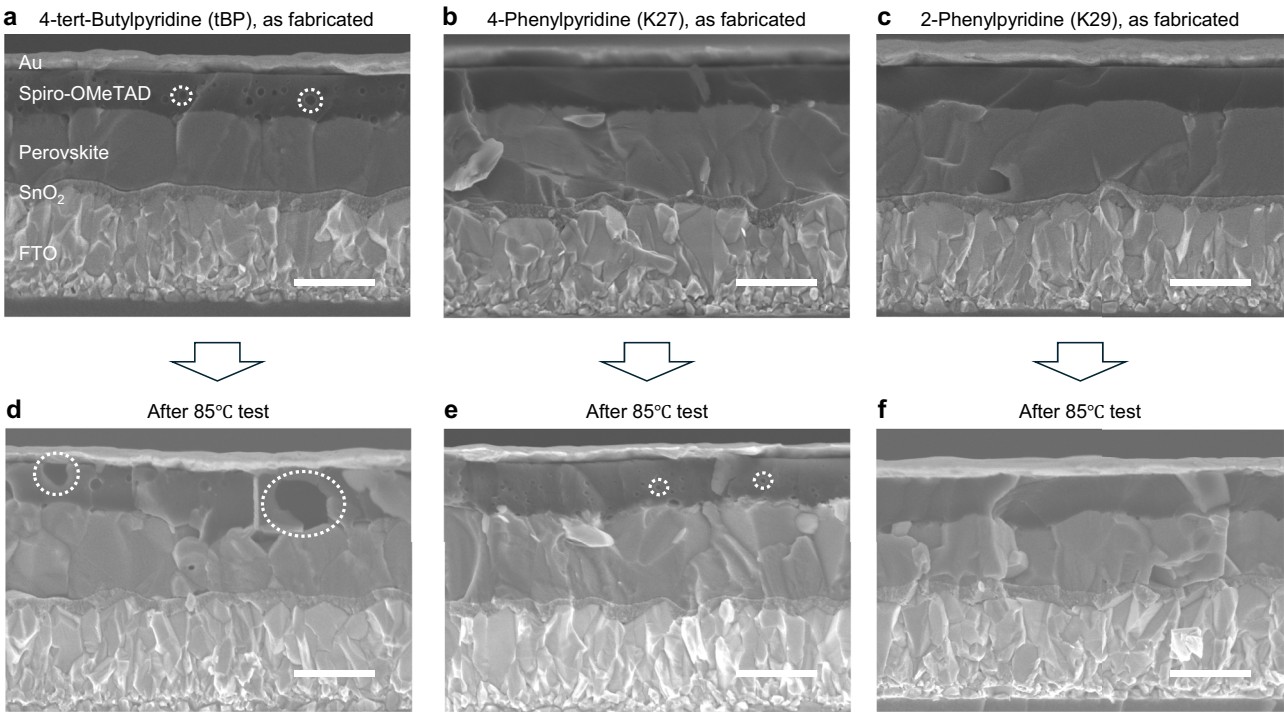

**Fig. 3 | SEM images of perovskite solar cells before and after 85 °C test. a-c**, SEM images of perovskite solar cells with 4-*tert*-butylpyridine (Li+tBP), 4-phenylpyridine (Li+K27), and 2-phenylpyridine (Li+K29), respectively. **d-f**, Those of SEM images after 400 hours of 85 °C test. Scale bar corresponds to 500 nm.

OMeTAD and perovskite layers. These small voids and/or by-products may contribute to the gradual decline in *FF* and *PCE* observed during the thermal stability test (Supplementary Fig. 45). Notably, the device with 2-phenylpyridine (Li+K29, HTL thickness >200 nm) demonstrated significant suppression of void formation in the Spiro-OMeTAD layer (Fig. 3f). Clear interfaces between the HTL/perovskite and HTL/Au layers were also observed. The boiling points are 274 °C for 3-phenylpyridine, 270 °C for 2-phenylpyridine, and 197 °C for 4-tert-butyl-pyridine, respectively. These values are consistent with the trends observed in Fig. 3. This void-free HTL facilitated by 2-phenylpyridine (Li+K29) can be responsible for the exceptional thermal stability of PSCs under prolonged thermal testing (Fig. 2b). Furthermore, we performed optical microscopy and AFM measurements of the HTLs before and after thermal aging (85 °C for 100 hours, Supplementary Figs. 73, 74). After thermal aging, the HTL with 4-tert-butylpyridine exhibited pin-hole formation and cracks originating from crystalline-like regions. These results are consistent with previous reports[96]. The depths of the pin-holes and cracks in the crystalline-like regions were in the range of 50–150 nm. Such defects are expected to deteriorate the electrical properties of the HTLs as well as the interfaces between HTL/Au and Perovskite/HTL. In contrast, the HTLs with 4-phenylpyridine and 2-phenylpyridine showed no significant morphological changes after the 85 °C test. These observations suggest that 4-phenylpyridine and 2-phenylpyridine effectively suppress pin-hole and crack formation in crystalline-like regions, which account for their higher thermal stability. We also examined the dissolution effect of the additives on lithium salts. Since the additives promote the dissolution of lithium salts and prevent the agglomeration of lithium salts, this may be a key factor affecting device performance and thermal stability. Lithium salts were dissolved in Spiro-OMeTAD solutions containing each additive. Supplementary Fig. 75 presents the relationship between thermal stability (PCE after 1000 h) and the solubility of lithium salts. tBP and 3-phenylpyridine (K28) dissolved 62.5 mg/mL of lithium salts, whereas 4-phenylpyridine (K27) and 2-phenylpyridine (K29) dissolved 55.6 mg/mL and 11.4 mg/mL, respectively. These results show no clear correlation between thermal stability and lithium salt solubility, suggesting that other mechanisms contribute to thermal stability.

The XRD patterns (Fig. 4a,b) were analyzed to evaluate the reactivity between the additives and the perovskite layer, comparing before and after 85°C tests. Samples were prepared same as the device structure of FTO glass/SnO$_2$/perovskite (PVK)/Spiro-OMeTAD(+ additives)/Au (Supplementary Fig. 76). Au layer was peeled out just before XRD measurement. Additives of 4-*tert*-butylpyridine (tBP), 4-phenylpyridine (K27), 3-phenylpyridine (K28), or 2-phenylpyridine (K29) were used for this experiment. For all the samples, peaks appeared at 13.9°, 19.8° and 24.3°, which are associated with (100), (110), and (111) of the perovskite crystal, respectively. Before 85 °C test (Fig. 4a), the sample with 4-phenylpyridine (K27) showed a small peak at 6.9° that could be from PbI$_2$-additives complex[91]. After 85 °C tests, samples with 4-tert-butylypyridine (tBP) and 4-phenylpyridine (K27) exhibit a clear peak at 6.9° that could be from PbI$_2$-additives complex. These are possibly harmful by-products leading to the instability of PSCs. It seems that these by-products themselves are not the primary reason for poor thermal stability, but it can be the secondary reason. Interestingly, samples with 2-phenylpyridine (K29) and 3-phenylpyridine (K28) appear no peak at 6.9°, but small peaks were observed at 12.6 corresponding to PbI$_2$. These results indicate that 2-phenylpyridine and 3-phenylpyridine have low reactivity with the perovskite layer. This lack of reactivity is attributed to the steric hindrance introduced by the meta and ortho substitutions, which can prevent coordination with PbI$_2$, thereby contributing to their excellent thermal stability (Fig. 2b).

The impact of additives on carrier dynamics was further examined using steady-state PL and time-resolved photoluminescence (trPL), as shown in Figs. 4c and 4d, respectively. Samples were prepared as glass/perovskite/Spiro-OMeTAD with each additive. The PL spectrum of the perovskite-only sample exhibited a prominent peak, indicating minimal non-radiative pathways, consistent with the extended carrier lifetime measured by trPL ($\tau_b$ = 5806.1 ns, Supplementary Table 62). In contrast, the sample with 4-*tert*-butylpyridine (PVK/tBP) showed quenched PL emissions and a reduced carrier lifetime ($\tau_b$ = 43.7 ns),

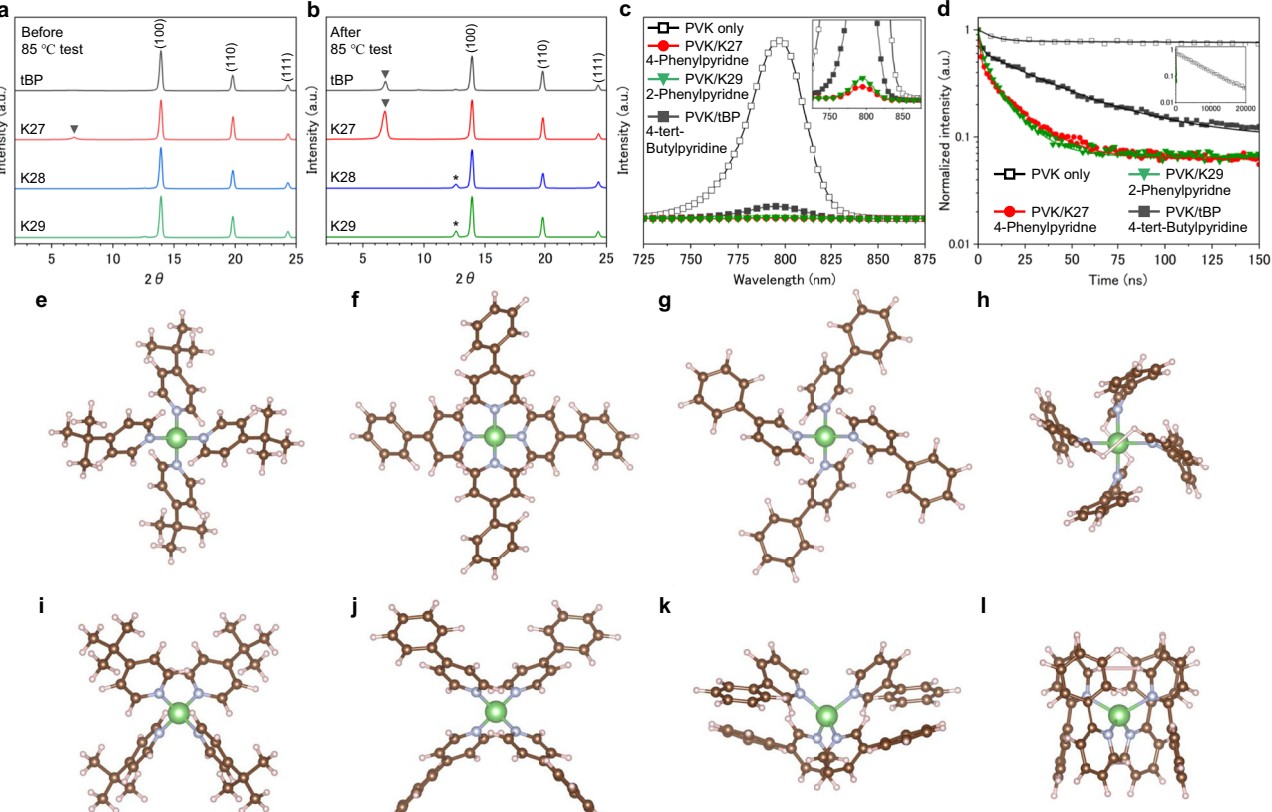

**Fig. 4 | Characterization for the reactivity and interaction of each additive.**
**a** XRD patterns of devices with each additive. **a** XRD patterns of samples with 4-tert-butylpyridine (tBP), 4-phenylpyridine (K27), 3-phenylpyridine (K28), and 2-phenylpyridine (K29). **b** those of XRD patterns after 85 °C aging for 100 hours. **c**, Steady-state PL spectra of samples. **d** Time-resolved PL spectra of samples. DFT calculated chemical structure with **e** 4- tert-butylpyridine (tBP) and lithium. **f** 4-Phenylpyridine (K27) and lithium. **g** 3-Phenylpyridine (K28) and lithium. **h** 2-Phenylpyridine (K29) and lithium. **i–l** Corresponding structures viewed from different angles.

attributed to charge transfer from the perovskite layer to the Spiro-OMeTAD layer. Notably, this quenching was significantly accelerated in samples incorporating 4-phenylpyridine (PVK/K27, $\tau_b = 18.5$ ns) and 2-phenylpyridine (PVK/K29, $\tau_b = 13.2$ ns). These results suggest enhanced charge transfer facilitated by these additives, potentially leading to improved photovoltaic performance in PSCs compared to tBP. The influence of this accelerated charge transfer on PSC performance is discussed in subsequent sections.

The interaction between additives and lithium was analyzed using DFT calculations (Fig. 4e-l). Figure 4e-h illustrate the coordination of 4-tert-butylpyridine (tBP) with lithium, 4-phenylpyridine (K27) with lithium, 3-phenylpyridine (K28) with lithium 2-phenylpyridine (K29), respectively, and Fig. 4i-l are corresponding structures viewed from different angles. All additives were found to form a tetrahedral arrangement as the most stable configuration, all with point group $S_4$. 4-tert-Butylpyridine (tBP) and 4-phenylpyridine (K27) were the similar geometries. Interestingly, 3-phenylpyridine exhibited a slightly more compact structure than 4-phenylpyridine (K27) due to geometric distortion induced by its molecular orientation. Furthermore, 2-phenylpyridine (K29) formed a significantly more compact coordination structure compared to the other additives, in which the lithium ion appeared to be tightly surrounded. The minimum edge lengths of the smallest boxes that could enclose each molecule were determined to be 10.93 Å for 4-tert-butylpyridine (tBP), 12.16 Å for 4-phenylpyridine (K27), 8.01 Å for 3-phenylpyridine (K28), and 9.06 Å for 2-phenylpyridine (K29), respectively. This difference in coordination may be related to the lower volatility and absence of voids observed in Spiro-OMeTAD with 2-phenylpyridine under the 85 °C test conditions.

When three molecules of each additive were coordinated with lithium, an energetically favorable state of 2-phenylpyridine, 3-phenylpyridine, and 4-phenylpyridine was point group $C_3$, while tBP was $D_3$, respectively. The desorption energy required to remove a coordinated molecule from lithium was calculated to be 0.667 eV for tBP, 0.659 eV for K27, 0.680 eV for K28, and 0.100 eV for K29, indicating that there is no relationship between thermal stability and ease of molecular desorption. Supplementary Figs. 77a, b depict the interactions between the coordinated additive-lithium complexes and Spiro-OMeTAD. The influence of π-interactions between the methoxyphenylethylamine group in Spiro-OMeTAD and the pyridine rings of the additives was evaluated. Calculations show that a positive formation energy is required for interaction between tBP and K27 and Spiro-OMeTAD to happen because steric effects require a large energy to deform Spiro-OMeTAD to allow such interactions to happen. These results are consistent with the X-ray photoelectron spectroscopy, which shows no significant chemical shift (Supplementary Fig. 78). These results suggest that π-interactions between the additives and Spiro-OMeTAD do not affect the overall stability or performance of the system.

We measured Fourier-transform infrared spectroscopy (FTIR) to investigate the molecular interactions (Supplementary Fig. 79). Peaks at 1061 cm$^{-1}$ and 1349 cm$^{-1}$ were observed, which correspond to the S=O stretching vibration of TFSI[63,101]. Interestingly, these peaks were shifted to higher wavenumber in the cases of 4-phenylpyridine and 2-phenylpyridine. This shift could be attributed to the stronger coordination of 4-phenylpyridine and 2-phenylpyridine with lithium, which reduces TFSI coordination and thereby strengthens the S=O bond.

For further understanding of the mechanism of the void-formation, we investigated the effect of additives on elemental distribution using ToF-SIMS (Supplementary Fig. 80). The measured samples were prepared with the configuration FTO/SnO$_2$/Perovskite/Spiro-OMeTAD (with additives)/Au, following the same procedure as device fabrication. The Au layer was peeled off immediately before the ToF-SIMS measurement. We compared 4-tert-butylpyridine and 2-phenylpyridine before and after thermal aging at 85 °C for 100 hours. Lithium migrated toward the ETL side and slightly accumulated at the surface, which is consistent with a previous report[89]. The lithium distribution was not significantly affected by the choice of dopant, nor by the thermal test. From these results, 2-phenylpyridine does not appear to promote a more homogeneous lithium distribution. For the fresh device with 4-tert-butylpyridine, the additive was mainly distributed in the Spiro-OMeTAD layer (Supplementary Fig. 80a). After the 85 °C test, however, 4-tert-butylpyridine had migrated substantially into the perovskite layer (Supplementary Fig. 80b). This migration is likely driven by its reaction with the perovskite, which is consistent with the by-product confirmed by XRD (Fig. 4b). In contrast, 2-phenylpyridine was mainly localized in the Spiro-OMeTAD layer, and its distribution showed no significant change after the 85 °C test (Supplementary Figs. 80c, d). This stability may contribute to its beneficial effect on thermal stability, probably because 2-phenylpyridine does not react with the perovskite and therefore does not diffuse into the perovskite layer. These ToF-SIMS results may explain the absence of voids in the 2-phenylpyridine-containing Spiro-OMeTAD layer after the 85 °C test. In the case of 4-tert-butylpyridine, its significant migration from Spiro-OMeTAD into the perovskite layer may have contributed to void formation within Spiro-OMeTAD (Supplementary Figs. 81a, b). By contrast, the void-free morphology observed with 2-phenylpyridine can be attributed to the stability of its distribution (Supplementary Fig. 81c, d).

**Photovoltaic performances with heteroaryl derivatives**

Additionally, to evaluate the impact of additives on photovoltaic performance, PSCs were optimized for high efficiency (Fig. 5a-e, Supplementary Figs. 82, 83, Supplementary Table 63). These optimizations included OAI passivation and a SnO$_2$ layer deposited via chemical bath deposition to enhance device performance. The PCE of devices incorporated with 4-phenylpyridine (Li+K27) was 23.2% which was higher than that of the tBP (22.4%), primarily due to improvements in both $V_{OC}$ and FF. Notably, devices incorporating 2-phenylpyridine (Li+K29) achieved a remarkable PCE of 25%, accompanied by an enhanced $V_{OC}$ of 1.165 V and FF of 0.82, with reduced hysteresis. These improvements are consistent with the fast-charge extraction measured by trPL (Fig. 4d). The exited carrier was extracted from the perovskite to HTL before undergoing carrier recombination, resulting in higher $V_{OC}$, FF, and PCE. This demonstrates the potential for achieving both high durability and conversion efficiency in the Spiro-OMeTAD system. We also examined whether the concentration of Spiro-OMeTAD influences device performance by comparing 30 mg/ml and 120 mg/ml solutions (Supplementary Fig. 84, 85). The results indicate that the higher concentration (120 mg/ml) improves reproducibility without significantly increasing resistance. UV–vis absorption spectroscopy was performed to evaluate the potential influence of the additives on the oxidation of the hole-transport layer (Supplementary Fig. 86). The UV–vis spectra showed no appreciable differences among the samples, suggesting that neither 4-phenylpyridine nor 2-phenylpyridine has a measurable effect on the oxidation process. Furthermore, energy level diagrams of Spiro-OMeTAD with additives were constructed based on photoelectron yield spectroscopy and Kelvin probe force microscopy (KPFM) (Supplementary Figs. 87–89). The valence band edge (Ev) of Spiro-OMeTAD with 4-tert-butylpyridine was −5.58 eV. In contrast, the

Ev values of Spiro-OMeTAD with 4-phenylpyridine, 3-phenylpyridine, and 2-phenylpyridine were −5.76 eV, −5.76 eV, and −5.83 eV, respectively, which are closer to that of perovskite (−5.89 eV) than the value obtained with 4-tert-butylpyridine. This alignment is likely to facilitate more efficient charge transfer, thereby contributing to improved photoconversion efficiency. The resistivity of Spiro-OMeTAD layers containing LiTFSI and each additive was 19,100 Ω cm for 4-tert-butylpyridine, 8540 Ω cm for 4-phenylpyridine, 6050 Ω cm for 3-phenylpyridine, and 2680 Ω cm for 2-phenylpyridine, respectively (Supplementary Fig. 90). These results suggest that 2,3,4-phenylpyridine has an advantage in the resistivity compared to 4-tert-butylpyridine, resulting in improved FF and PCE. Furthermore, mini modules incorporating 4-phenylpyridine achieved a $PCE_{active\ area}$ of 21.2% ($PCE_{apature\ area}$ = 18.3% with 1.664 cm$^2$ of aperture area), indicating that this strategy is scalable to larger device areas. Outdoor performance testing (Fig. 5f and 5g) revealed stable device operation with 2-phenylpyridine (Li+K29), with minor distortions in performance curves on e.g. 3$^{rd}$ of January due to intermittent cloud cover (Supplementary Fig. 91). In the case of tBP, the maximum power point voltage of the device continuously decreased to 81% of its initial value after 1,248 hours (Supplementary Fig. 92). As for 2-phenylpyridine, maximum power point voltage dropped to about 90% of its initial value within 300 hours from the start of the experiment, and no further voltage drop occurred for the till 1570 hours (90% of maximum power point voltage at 1570 hours). Importantly, 94% of output power was observed from the initial power after 1570 hours of outdoor testing, which exhibits remarkable light and cycle stability. From the above experiment results, it was found that additives that improve the thermal stability of PSCs have superior characteristics to conventional materials in terms of conversion efficiency and outdoor durability.

In conclusion, we demonstrated that incorporating additives into Spiro-OMeTAD can effectively address the thermal stability issue. Our key findings are that the heteroaryl derivatives with phenyl group control void formation in the Spiro-OMeTAD layer and minimize reactivity with the perovskite layer, significantly enhancing the thermal stability of PSCs under 85 °C conditions. The PSCs incorporated with 3-phenylpyridine (Li+K28) and 2-phenylpyridine (Li+K29) maintained 101% and 104% of their initial PCE after 2,400 hours of 85 °C test, respectively, showcasing their remarkable durability. Additionally, the additives improved charge extraction, contributing to enhanced photovoltaic performance with a PCE of 25%. Also, the outdoor test shows 90% of the initial power point voltage after 1570 hours with continuous MPPT measurement, showing remarkable light and cycle stability. A detailed investigation of 38 heteroaryl derivatives and 60 additive combinations reveals the underlying mechanisms that enable these improvements, offering a pathway toward thermally stable, light-stable, and high-performance PSCs with thick HTL ( > 200 nm), which is suitable for scalable fabrication with blade coating.

## Methods
### Materials
Cesium iodide, lead iodide, formamidinium iodide, heteroaryl derivatives, tin (IV) chloride, and thioglycolic acid were purchased from TCI. n-octylammonium iodide was purchased from GreatCellSolar. Urea was purchased from Fujifilm Wako. The FTO glass substrates of TEC-10 and VU-glass were purchased from Nippon Sheet Glass and AGC, respectively. Chlorobenzene, bis(trifluoromethane)sulfonimide lithium salt, rubidium iodide, and the cobalt-complex (FK209) were purchased from Aldrich. DMF, DMSO, isopropanol, and acetonitrile were purchased from Aldrich. Methylammonium chloride was purchased from Xi'an Yuri Solar. SnO$_2$ nanoparticle dispersed solution (S-8) was purchased from Taki Chemical. Anti-reflection film (Mosmite) was provided by Mitsubishi Chemical Group. All chemicals were used as received without further purification.

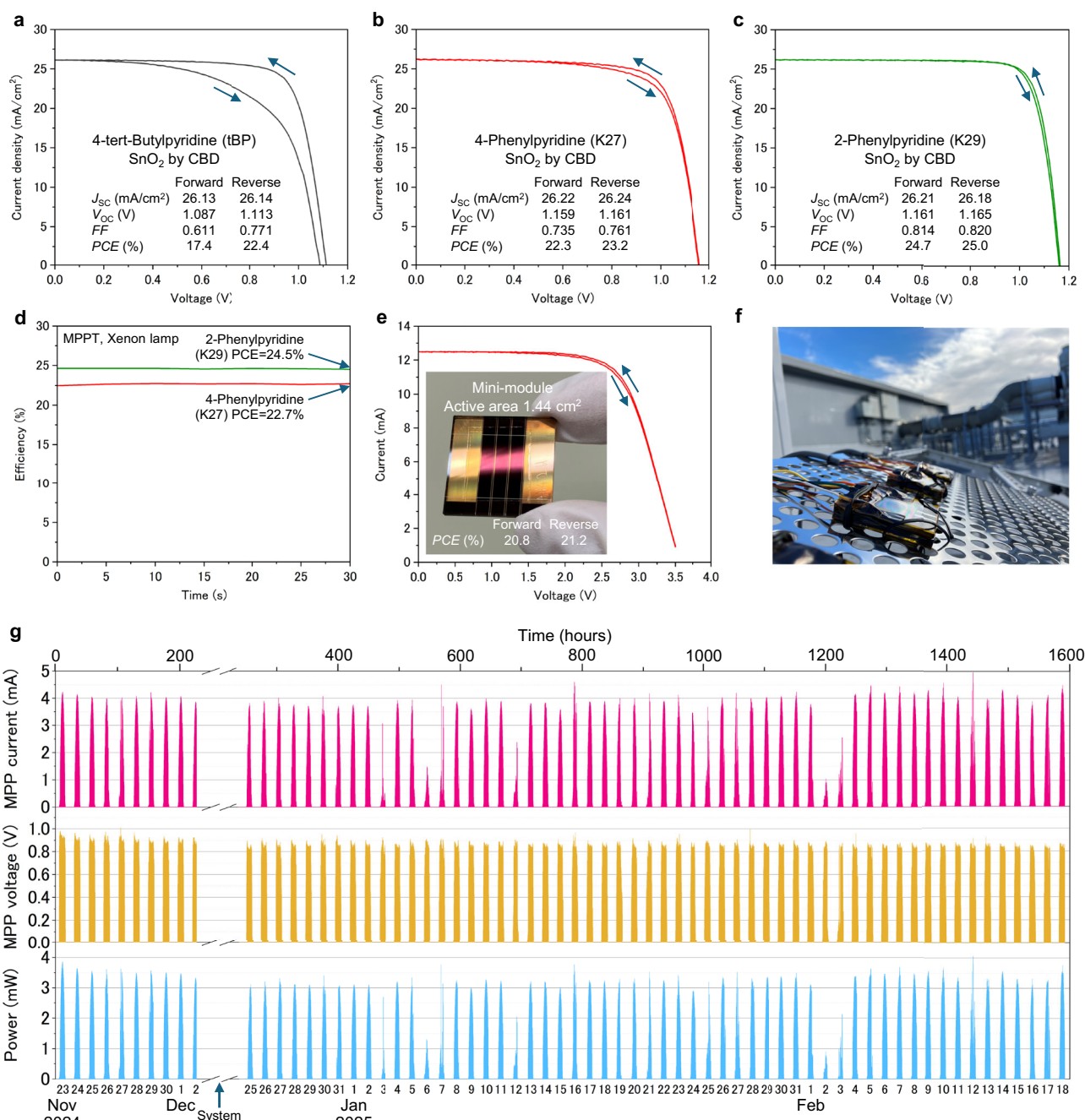

**Fig. 5 | Photovoltaic performances of the perovskite solar cells with each additive and outdoor stability. a–c** *I-V* scans of the perovskite solar cells with 4-*tert*-butylpyridine (Li+tBP), 4-phenylpyridine (Li+K27), and 2-phenylpyridine (Li+K29). **d** MPP tracking of devices with 4-phenylpyridine (Li+K27), and 2-phenylpyridine (Li+K29), **e** *I-V* scans of mini-module device. **f** Devices for outside stability. **g** MPP tracking data of outside devices with 2-phenylpyridine (Li+K29) as a function of days.

## Device fabrication

Encapsulated PSCs were fabricated by the following steps. Devices were fabricated under ambient dry air (dew point = −25 - −40 °C). The FTO glass (TEC-10) was UV/O₃ treated for 30 minutes. The FTO glass was etched by laser. Then, 200 μL of the SnO₂ dispersed solution (S-8) was spin-coated on the top of the FTO substrate at 3000 rpm for 20 s. These substrates were annealed at 100 °C for 5 min and 150 °C for 30 min sequentially. For the perovskite deposition, a perovskite precursor composed of CsFAPbI₃ as prepared by dissolving 556.7 mg of lead iodide, 180.6 mg of formamidinium iodide, 13.6 mg of cesium iodide and 7.1 mg of methylammonium chloride in a solution of DMSO (300 μL) and DMF (700 μL). The Spiro-OMeTAD precursor solution

was prepared by dissolving 120 mg of Spiro-OMeTAD in 1000 μL chlorobenzene, 70.2 μL 4-*tert*-butylpyridine, 41.4 μL of bis(tri-fluoromethane)sulfonimide lithium in acetonitrile solution (99 mg of LiTFSI + 263 μL of acetonitrile) and/or 32.4 μL of tris(2(1H-pyrazol-1-yl) −4-*tert*-butylpyridine)cobalt(III) tri[bis(trifluoromethane)sulfonimide] in acetonitrile solution (196 mg of LiTFSI + 379 μL of acetonitrile). In the case of other heteroaryl derivatives, the same molar amount was added to the Spiro-OMeTAD solution instead of 4-*tert*-butylpyridine. Thereafter, two-step spin-coating of perovskite precursor was accomplished through a two-step program running at 0 rpm (loading for solution), 1000 rpm, and 6000 rpm for 10 s, 10 s (+2 s acceleration), and 30 s (+2 s acceleration), respectively. At 48 s of the spin-

coating program, 200 µL of chlorobenzene was dispensed. The substrates were then transferred for annealing at 140 °C for 5 min. Then, hole transport layer was spin-coated (4000 rpm, 20 s) onto the perovskite layer then transferred for annealing at 85 °C for 10 min for dying. Laser scribing was performed to remove the unnecessary perovskite layer and to make contact. Finally, a 70 nm gold contact layer was deposited by a thermal evaporation method. The devices were encapsulated with a glass cap, UV-glue, and getter seal.

For high-efficiency PSCs, VU glass was used instead of the TEC-10. The chemical bath deposition (CBD) method was used for $SnO_2$ deposition instead of $SnO_2$ nano-particle (S-8). CBD solution was prepared with 1100 mg of tin (II) chloride, 5000 mg of urea, 100 µL of thioglycolic acid, 5000 µL of hydrochloric acid and 400 mL of deionized water. Substrates were dipped in the CBD solution and annealed at 90 °C for 5 hours, followed by sonification with deionized water for 5 min and isopropanol for 5 min. Substrates were annealed at 170 °C for 1 hour, then $UV/O_3$ cleaned for 30 min. For the perovskite deposition, a perovskite precursor composed of $CsFAPbI_3$ as described before or $RbCsFAPbI_3$ prepared by dissolving 594.0 mg of lead iodide, 180.6 mg of formamidinium iodide, 13.6 mg of cesium iodide, 7.1 mg of methylammonium chloride, 6.7 mg of rubidium iodide in a solution of DMSO (300 µL) and DMF (700 µL). After perovskite deposition, 9.8 mg of *n*-octylammonium iodide (OAI) in 1 mL of isopropanol was spin-coated dynamically on the perovskite for 20 seconds at 4000 rpm. IPA (50 µL) was spun on samples (4000 rpm, dynamic) for stability measurement devices. Then samples were annealed at 100 °C for 5 min. After Spiro-OMeTAD deposition, samples were annealed at 50 °C. After Au deposition, an anti-reflection film was attached to the devices. Other procedures (e.g., Spiro-OMeTAD solution, spin-coating program) are the same as encapsulated PSCs fabrication as described above. Note that regarding devices with tBP, the MACl concentration of the perovskite layer was increased from 10% to 20%, which is an optimized condition for tBP to achieve high efficiency.

For the mini-module fabrication, a laser scribing system was used for P1, P2, and P3 scribing. The width of P1, P2, and P3 was 100 µm, and the margin between P1-P2 and P2-P3 was 100 µm. Pulse width was 1 picosecond. The wavelength of the laser was 1030 nm. Butylammonium iodide + octylammonium iodide (BAI + OAI) passivation was incorporated according to a previous report[97]. 4-Phenylpyridine was used as an additive in Spiro-OMeTAD.

Devices for the outside were prepared with encapsulated PSCs. 2-Phenylpyridine was used as an additive in Spiro-OMeTAD. OAI passivation was incorporated. After the fabrication of PSCs, devices were stored at 85 °C for 40–50 hours for stabilization purposes. Butyl seal was used for additional encapsulation.

### Characterization

AM1.5 pseudo-solar light (100 mW cm$^{-2}$) was generated using a solar simulator equipped with a xenon lamp (Bunkou Keiki). Standard silicon solar cells (BS-521BK, Bunkou Keiki) were used to calibrate the incident light intensity. The *I–V* measurements were performed by using a solar simulator, carried out from 1.2 to 0 V and from 0 to 1.2 V as reverse and forward scans, respectively, using a mask of 0.0887 cm$^2$. The scanning step and speed for solar cells were 10 mV and 100 mV s$^{-1}$, respectively. The scanning step and speed for mini modules were 50 mV and 500 mV s$^{-1}$, respectively. IV curves of devices with or without encapsulation were measured at room temperature in air. Thermal stability tests were performed for encapsulated devices in dark in air without voltage bias. Samples were kept in the electrical oven at 85 °C. Preconditioning of the device was not applied before the measurement. The active area and aperture area for mini modules were 1.44 cm$^2$ and 1.664 cm$^2$, respectively. The active and aperture area for mini modules was determined by the active area excluding or including P1, P2, and P3 area, respectively, and mini modules were measured

without a mask. SEM images were obtained using a field emission scanning electron microscope (HITACH corporation, SU9000). X-ray diffraction (XRD) analysis was carried out using a D8 Advance diffractometer (Rigaku, SmartLab) with Cu Kα radiation (λ = 1.5418 Å). Samples for PL measurement were prepared with glass/perovskite/additives. Additives were deposited by spin-coating (4000 rpm for 20 s) with the additive solution (70.2 µL of 4-*tert*-butylpyridine in 300 µL of chlorobenzene) on the perovskite layer. In the case of 4-phenylpyridine, 3-phenylpyridine, or 2-phenylpyridine samples, the same molar amount of 4-*tert*-butylpyridine was added in 300 µL of chlorobenzene. Substrates were annealed at 85 °C for 10 min. The PL lifetimes were measured using a time-correlated single photon counting system (Fluorolog-QM, Horiba) with an excitation at 634 nm. The PL emission was measured with an excitation at 640 nm. Samples for PL measurement were prepared with glass/perovskite/Spiro-OMeTAD with additives. Samples were annealed at 85 °C for 10 min. X-ray photoelectron spectroscopy (XPS) was performed by ULVAC-PHI PHI5000 VersaProbe. Samples for XPS were prepared as FTO/$SnO_2$/perovskite/Spiro-OMeTAD with additives. Samples were annealed at 85 °C for 10 min. IPCE measurement was performed by CPE-2000 (Bunkoukeiki Co. Ltd). Samples were prepared with anti-reflection film. Outside performance was measured by the maximum-power point tracking system (24-channel µMPPT, Laboratory of Photovoltaics and Optoelectronics (LPVO)). The active area of the device was 5 mm×5 mm. Samples were set on the outside stage without the mask. The angle of the outside stage was 20 degrees and faced south. Samples were connected by a 4-terminal-method. Short-term MPPT under the solar simulator was measured by the MPPT system by SYSTEMHOUSE SUNRISE. UV-vis was measured by MSV-5800 Microscopic Spectrophotometer (JASCO). Depth profiles of ToF-SIMS were performed by Toray Research Center with a GCIB beam with sample cooling. Resistivity was measured with the van der Pauw method. The Spiro-OMeTAD layers were deposited on glass substrates by spin-coating. Spiro-OMeTAD solutions were prepared using the same procedure as in the device fabrication. FTIR was measured by IRT-5200 Irtron Infrared Microscope (JASCO). Photoelectron yield spectroscopy (PYS) was measured by an ionization energy measurement system (BIP-KV100-WB, Bunkoukeiki). AFM/KPFM measurements were performed using an AFM100 (HITACHI). A rhodium-coated cantilever tip was employed, with an associated work function of 5.4 eV. The optical microscope was measured by the laser microscope VK-X3100 (KEYENCE).

### Calculations

Calculations were conducted using the Gaussian16 code. The B3LYP/6-31 + + G(d,p) basis set was used with the Opt=Tight convergence criterion. The net electric charge was +1, and the spin multiplicity was 1.

### Reporting summary

Further information on research design is available in the Nature Portfolio Reporting Summary linked to this article.

## Data availability

The data of this study are provided in the supplementary information. Additional data can be available from the corresponding author on request.

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

## Acknowledgements

This article is based on results obtained from a project, Grant-in-Aid for Research Activity Start-up (23K19275) from the Japan Society for the Promotion of Science (JSPS) and JPNP21016 commissioned by the New Energy and Industrial Technology Development Organization (NEDO).

## Author contributions

H.K. conceived and conceptualized ideas for this study and contributed to device fabrication, investigation, project administration, formal analysis, writing of the original draft, and funding acquisition. S.M. contributed to device fabrication. N.E. contributed to the SEM measurement. N.N. contributed to PL measurement. Y.H. contributed to DFT calculations. K.Y., M.Y., K.T., H.T., and A.N. contributed to laser scribing for the mini module. T.K. contributed to the conductivity measurement. T.N.M. contributed to supervising and funding acquisition. All authors contributed to writing, reviewing, and editing.

## Competing interests

The authors declare no competing interests.
