## [Transparent Peer Review file · Nature Communications]

Heteroaryl Derivatives for Hole-Transport Layers Improve Thermal Stability of Perovskite Solar Cells

Corresponding Author: Dr Hiroyuki Kanda

Version 0:

Reviewer comments:

Reviewer #1

(Remarks to the Author)

The manuscript presents a comprehensive study on the incorporation of heteroaryl derivatives as additives in Spiro-OMeTAD to enhance the thermal stability and photovoltaic performance of perovskite solar cells (PSCs). The work is well-structured and tackles a crucial challenge in the field of PSCs, namely, thermal instability. The experimental findings are promising, particularly regarding the remarkable thermal stability achieved with 3-phenylpyridine and 2-phenylpyridine additives. However, several critical issues must be addressed to improve the clarity, rigor, and overall impact of the study. I recommend major revisions before the manuscript can be considered for publication.

1. Selection Criteria for Heteroaryl Derivatives

The study investigates 38 heteroaryl derivatives; however, the rationale behind the selection of these specific compounds is not clearly justified. A discussion on the selection criteria—such as volatility, steric hindrance, electronic properties, or previous literature reports—would provide valuable context and strengthen the study's foundation.

2. Fill Factor (FF) Discrepancy

The device with a passivation layer and 4-tert-butylpyridine (tBP) as an additive exhibits an FF below 70%, which appears significantly lower than commonly reported values. The authors should provide an explanation for this discrepancy.

3. Comparison of Stability Among Additives

The claim that A81 (3-Phenylpyridine) and A82 (2-Phenylpyridine) offer superior stability compared to A15 (4-Phenylpyridine) is not convincingly demonstrated, as the initial efficiencies of these devices differ significantly. From an absolute efficiency standpoint, A15 may contribute to greater overall energy output during aging. This needs to be addressed with additional discussion or data normalization.

4. Void Formation in the HTL

The manuscript attributes void formation in the hole transport layer (HTL) to the use of low-boiling-point additives such as 4-tert-butylpyridine, which is reasonable. However, the observation that A82 (2-Phenylpyridine) does not induce void formation, while A15 (4-Phenylpyridine) does, requires further explanation. The authors should clarify whether they have investigated the thermodynamic properties of complexes formed between these additives and LiTFSI. Additionally, have they measured the glass transition temperature (T_g) of HTLs containing different additives? This data could provide insight into differences in void formation behavior.

5. Reactivity Between Perovskite and Pyridine Derivatives

The study reports that spin-coating pyridine derivatives directly onto perovskite films leads to degradation. However, in standard device fabrication, spin-coating HTL solutions containing 4-tert-butylpyridine does not appear to significantly impact device performance. The authors should clarify whether they observed the emergence of new XRD diffraction peaks in aged devices, similar to those found in direct pyridine spin-coating experiments. Furthermore, the claim that steric hindrance differences between 4-phenylpyridine and 3-phenylpyridine significantly impact coordination behavior is not well-supported. The authors should provide further discussion or computational evidence to substantiate this point.

6. Typographical Errors

Minor typographical errors exist throughout the manuscript, such as "phyenylpyridine" instead of "phenylpyridine" in the Figure 3 captions. A thorough proofreading is recommended to improve readability and ensure accuracy.

Reviewer #2

(Remarks to the Author)

The authors have done a substantial amount of work comparing the effects of different tBP substitutes on improving the

thermal stability of doped Spiro-OMeTAD. This is a commendable effort. Although the author provides extensive thermal stability test data demonstrating the enhancement of device thermal stability with 3-phenylpyridine and 2-phenylpyridine, the explanation of the mechanism is quite lacking. More characterisation and analysis are needed to substantiate the role of 3-phenylpyridine and 2-phenylpyridine. Hope the following questions can be addressed in the revised manuscript.

1. For the thermal stability, SnO₂ dispersed solution was used to fabricate SnO₂ carrier-transport-layer and CsFAPbI₃ was used to fabricate perovskite, whereas for the efficiency part, the carrier-transport-layer was changed to CBD SnO₂, and the perovskite was changed to RbCsFAPbI₃ with excess PbI₂ in precursor. It is necessary to provide the stability comparison data of highly efficient PSCs (dispersed-solution-SnO₂ vs CBD, CsFAPbI₃ vs RbCsFAPbI₃ with excess PbI₂).
2. It is necessary to provide the IPCE data and integrated current density of highly efficient PSCs shown in Fig.5c.
3. Solely using DFT to confirm the interaction between additives, lithium and Spiro-OMeTAD is not sufficient. Fourier-transform infrared spectroscopy (FTIR) is needed to evidence the presence of interaction.
4. Traditional Spiro-OMeTAD HTL suffers crystallization after thermal aging which worsens the electrical properties of HTL. Therefore, it is necessary to provide AFM and microscope photographs of the HTLs before and after thermal aging.
5. Additives could influence the oxidation of hole-transport layer. In this work, does the best two additives influence the oxidation/doping progress of hole-transport layer? Accelerate or postpone it? It is necessary to provide UV-vis absorption data to analyse the HTL before and after oxidation. UPS/KPFM is also necessary to investigate the associated working function/energy level changes.
6. Lithium salt was observed to accumulate at interfaces (Perovskite solar cells based on spiro-OMeTAD stabilized with an alkylthiol additive Nature Photonics, 2023, 17, 96-105). Are the additives proposed in this work helpful to obtain homogenous dopant distribution? Also, it is important to locate the position of additives. Is it homogeneously distributed throughout the HTL? Therefore, time-of-flight secondary-ion mass spectrometry (TOF-SIMS) or other characterization methods which can clearly demonstrate the out-of-plane component distribution is important to help readers understand the benefits of new the additives proposed in this work.
7. The poor thermal stability of doped Spiro-OMeTAD is not solely attributed to the additives (such as Li, tBP); it may also be related to intrinsic factors of Spiro-OMeTAD itself. For instance, its low T_g makes it susceptible to molecular segmental motion, and non-crystalline Spiro-OMeTAD can undergo local recrystallization at high temperatures. Additionally, the diffusion of I⁻ from the perovskite into HTL can also lead to the failure of Spiro-OMeTAD. Given this, why does the substitution of tBP with 3-phenylpyridine and 2-phenylpyridine result in a significant improvement in thermal stability? Please provide more evidence to determine whether 3-phenylpyridine and 2-phenylpyridine influence the factors mentioned above.
8. What are the boiling points of 3-phenylpyridine, 2-phenylpyridine, and tBP? Could the differences in the results shown in Figure 3 be due to the different boiling points of the three materials?
9. Please provide the AFM images of Spiro-OMeTAD doped with the three additives, Li+3-phenylpyridine, Li+2-phenylpyridine, and Li+tBP, before and after heating, in order to observe the behavior of Li⁺ before and after heating.
10. For stability, the thickness of Spiro-OMeTAD (120 mg/ml) has been increased in this manuscript. However, thicker Spiro-OMeTAD tends to increase the resistance of the HTL layer. Any trade-off benefit here?

Reviewer #3

(Remarks to the Author)

In this manuscript, the authors consider tBP as the key factor for the instability of Spiro-OMeTAD-based devices at high temperatures. They employed a large number of heteroaryl derivatives to verify which materials could improve thermal stability. Finally, the authors proposed that three types of derivatives have a positive effect on thermal stability, and based on this, devices were fabricated, which showed good stability in outdoor stability tests. Overall, the authors are thanked for their extensive work and experiments in screening suitable derivatives, and for providing reliable efficiency aging data instead of the normalized data in most literature. The authors also provided some mechanistic explanations to better understand the impact of such additives on the stability of Spiro-OMeTAD. However, despite the numerous experiments and tests conducted by the authors, the research results are not persuasive enough, and the mechanistic explanations lack new ideas, failing to arouse greater interest among readers. In addition, there are some logical issues in this research. Therefore, I do not recommend further consideration of this manuscript for publication in Nature Communications. The specific comments are as follows:

Major comments:

1. The authors believe that the thermal instability of tBP stems from its volatilization at high temperatures, which causes pore problems in Spiro-OMeTAD, thereby leading to the thermal instability of the devices. However, for Spiro-OMeTAD-based devices, the interaction between tBP and Li salts cannot be ignored. The core function of tBP doping is to promote the dissolution of lithium salts and prevent the agglomeration of lithium salts. Nevertheless, in this study, the authors used a large number of additives without considering the dissolution effect of the materials on lithium salts. This may be a key factor affecting device performance and thermal stability, rather than the effect brought about by the structure of the derivatives themselves. The authors did not rule out this key factor in their own research.
 2. Although the author provided a lot of device aging data for different additives in the supporting information, the device performance indicated by these data is really too low. Five years ago, it was very easy for devices using TBP to achieve an efficiency of over 20%, and it was not difficult to reach an efficiency of over 23% after passivation. However, the author only obtained an efficiency of around 15%, which indicates that there are significant problems with the perovskite thin film itself. The initial efficiency of the additives A80 and A81 proposed by the author is only above 14%, and using such data to measure the thermal stability of the devices lacks sufficient persuasiveness. In addition, after the modification of the electron transport layer (ETL) and passivation, the efficiency can be increased to 25%, but the final efficiency of the device modified with TBP is only 20.5%, which is hardly convincing.
- It is insufficient to merely use TRPL to characterize the performance of the Spiro-OMeTAD thin films after the addition of

different additives. If the modified Spiro-OMeTAD has sufficient performance advantages, the author should provide more characteristics of the optimized Spiro-OMeTAD, including the energy band, electrical conductivity, and so on.

3. The author spent a great deal of space in the previous part of the manuscript on using additives to address the thermal instability of perovskite devices. However, in the subsequent characterization of the optimized devices, the author tested the operational stability under actual outdoor ambient. Although this outdoor stability is crucial for perovskite devices, I fail to find sufficient logical coherence between the two parts. According to the temperature diagram provided by the author in the SI, it seems that the stability at high temperatures is not covered. Instead, the author emphasizes the performance under diurnal cycles. Therefore, I believe there are significant logical issues in this section. The author should focus on and provide the thermal stability performance when the efficiency is optimized to over 25%.

Other comments:

--Table 1 shows the code of different additives, which is not necessary to present it in main text.

--What is the author's numbering criterion for the additives? It seems rather chaotic in the article, as there is no specific order and some of the numbers are missing.

--In fig.3, the SEM images of perovskite film based on different spiro-OMeTAD show different thickness, the authors should confirm this.

--TRPL curves are very strange compared to other literature, the authors should re-test them.

--From the EQE curves and integrated Jsc in Fig.S69, the PCE of devices in Fig.5a-c should not obtain such high Jsc. The performance of these devices is overestimated.

--It is rarely that 9.8 mg/mL OAI was used for passivating the perovskite surface because it is hydrophobic to influence the deposition of Spiro-OMeTAD, the authors should confirm this.

Reviewer #4

(Remarks to the Author)

Version 1:

Reviewer comments:

Reviewer #1

(Remarks to the Author)

The authors have added substantial new data that more convincingly support the earlier conclusions, and they have addressed the majority of my previous questions satisfactorily. The manuscript has improved considerably.

1, The attempt to determine the thin-film T_g by DSC was unsuccessful. Beyond DSC, there are several film-compatible methods to measure the glass transition temperature, such as variable-temperature spectroscopic ellipsometry (identifying the slope change in refractive index/thickness vs temperature) and dynamic mechanical analysis (DMA). Please consider adopting at least one of these approaches and report key parameters (heating rate, substrate, film thickness, etc.).

2, On page 21, the manuscript states: "Interestingly, 3-phenylpyridine exhibited a slightly more compact structure than 4-phenylpyridine (A15) due to geometric distortion induced by its molecular orientation." At present this "more compact/crowded" description is based on visual inspection, which is somewhat qualitative. I recommend quantifying the interaction/coordination strength (e.g., binding energy or dissociation barrier from calculations or experiments) to better substantiate the claimed tendency toward dissociation, and include methods and uncertainties in the main text.

3, The compound codes have been revised, but a few instances of the old codes remain. Please perform a thorough consistency check across the entire manuscript, including figure captions, tables, Supporting Information, and cross-references.

Reviewer #2

(Remarks to the Author)

The authors have made detailed response to the reviewers' comments. The revision is greatly improved in the current form. I think this work is suitable for publication in NC.

Reviewer #3

(Remarks to the Author)

it can be accepted.

Reviewer #4

(Remarks to the Author)

Version 2:

Reviewer comments:

Reviewer #1

(Remarks to the Author)

I would suggest to accept the ms as is.

Reviewer #4

(Remarks to the Author)

Response to reviewer

We appreciate so much for the effort and dedicated work of the editor and the reviewers. We are grateful for offering the opportunity to improve our work as major revision. We have judiciously taken all comments into account following the comments raised.

Comments on the reviewers: **in red color**

Our response: **in blue color**

Revised manuscript: **in yellow highlighted**

Reviewer #1 (Remarks to the Author):

Recommendation: Major revisions

The manuscript presents a comprehensive study on the incorporation of heteroaryl derivatives as additives in Spiro-OMeTAD to enhance the thermal stability and photovoltaic performance of perovskite solar cells (PSCs). The work is well-structured and tackles a crucial challenge in the field of PSCs, namely, thermal instability. The experimental findings are promising, particularly regarding the remarkable thermal stability achieved with 3-phenylpyridine and 2-phenylpyridine additives. However, several critical issues must be addressed to improve the clarity, rigor, and overall impact of the study. I recommend major revisions before the manuscript can be considered for publication.

Answer: Thank you so much for your valuable comments and recommendations for major revision. We revised manuscript according to reviewer's pointing.

1-1. Selection Criteria for Heteroaryl Derivatives

The study investigates 38 heteroaryl derivatives; however, the rationale behind the selection of these specific compounds is not clearly justified. A discussion on the selection criteria—such as volatility, steric hindrance, electronic properties, or previous literature reports—would provide valuable context and strengthen the study’s foundation.

Answer: Thank you for your valuable comments. We revised manuscript by adding an explanation of the rationale behind of the selection of these specific compounds referring previous literature reports.

Sentence below was added on page 6.

It is desirable to examine the selection of heteroaryl derivatives and their impact on durability from the following perspectives.

For the volatility, some literature insists on the volatility of the tBP which can be the reason for the weak durability of the Spiro-OMeTAD⁹⁶. Thus, it is important to evaluate heteroaryl derivatives with different boiling points to investigate the effects of volatility on the thermal stability. For example, boiling points of 2-phenylpyridine, 3-phenylpyridine, and 4-phenylpyridine corresponds to 270 °C, 274 °C, and 281 °C, respectively, which is higher than tBP (197 °C). Also, derivatives with lower boiling point should be selected for comparison e.g. 4-ethylpyridine (165 °C), 4-aminopyridine (157~162 °C).

Regarding steric hindrance, Y. Yue et al., introduced 2-amylpyridine instead of tBP, which is substituted with an amyl group at ortho position⁹¹. They explained that the steric hindrance from the ortho-substitution of pyridine ring may prevent the formation of complex with additives and perovskite, which could increase the durability of perovskite solar cells. Therefore, we can select ortho, meta, and para substituted heteroaryl derivatives to investigate the effect of the steric hinderance on the durability.

As for the electronic properties, they can be tuned by introducing various functional groups and heteroatoms onto the hetero ring. These functional groups exhibit electron-donating or electron-withdrawing effects, which influence the electronegativity of the pyridine ring and its

interaction with lithium. Evaluating these effects is important to determine whether the electronic state influences thermal stability. Therefore, a variety of functional groups with different electron-donating or -withdrawing characteristics, as well as heteroatoms, need to be verified.

1-2. Fill Factor (FF) Discrepancy

The device with a passivation layer and 4-tert-butylpyridine (tBP) as an additive exhibits an FF below 70%, which appears significantly lower than commonly reported values. The authors should provide an explanation for this discrepancy.

Answer: Thank you for your pointing. The reason for the discrepancy is that the optimal condition of the perovskite solar cells can be different by different additives. To improve the FF of the perovskite solar cells with tBP, we optimized device perovskite recipe by increasing MACl concentration from 10% to 20%. By this optimization, devices with tBP achieved 0.771 of FF and 22.4% of PCE. These results were added to the manuscript.

Fig. 5 a $I-V$ scans of the perovskite solar cells with 4-tert-butylpyridine (Li+tBP). Note that MACl concentration increased from 10% to 20% to improve efficiency for devices with tBP.

Table S63. Photovoltaic parameters for perovskite solar cells with tBP. MAI concentration increased from 10% to 20% to improve efficiency.

Cell	Jsc	Jsc	Voc (V)	Voc (V)	FF (-)	FF (-)	PCE (%)	PCE (%)
	(mA/sq)	(mA/sq)						
	Forward	Reverse	Forward	Reverse	Forward	Reverse	Forward	Reverse
1	26.17	26.17	1.058	1.085	0.549	0.726	15.21	20.63
2	26.13	26.13	1.087	1.113	0.612	0.771	17.37	22.43
3	26.20	26.19	1.103	1.123	0.541	0.717	15.66	21.11
4	26.21	26.21	1.107	1.125	0.455	0.707	13.20	20.86

Sentence below was added on page 32.

Note that regarding devices with tBP, MAI concentration of the perovskite layer was increased from 10% to 20%, which is optimized condition for tBP to achieve high efficiency.

1-3. Comparison of Stability Among Additives

The claim that A81 (3-Phenylpyridine) and A82 (2-Phenylpyridine) offer superior stability compared to A15 (4-Phenylpyridine) is not convincingly demonstrated, as the initial efficiencies of these devices differ significantly. From an absolute efficiency standpoint, A15 may contribute to greater overall energy output during aging. This needs to be addressed with additional discussion or data normalization.

Answer: Thank you for your valuable comments. We agree that claiming 3-phenylpyridine and 2-phenylpyridine are superior stability compared to 4-phenylpyridine is an overstatement. We added discussion considering energy output during aging as well as added normalized data.

The sentence below was added on page 12.

4-Phenylpyridine showed higher PCE than 2-phenylpyridine and 3-phenylpyridine for approximately 1100 hours from the start of the 85 °C stability test. This suggests that 4-phenylpyridine has an advantage in initial PCE, which may contribute to greater overall energy output during the degradation period. For example, if 1000 hours accelerated test at 85°C is sufficient to ensure practical outdoor durability, 4-phenylpyridine could be more advantageous in practical use compared to 2-phenylpyridine and 3-phenylpyridine. Therefore, it is premature to conclude which molecular orientation is the most superior.

Figure S70. Normalized 85 °C thermal stability result with 4-phenylpyridine (Li+A15), 3-phenylpyridine (Li+A81), 2-phenylpyridine (Li+A82), and 4-tert-butylpyridine (Li+tBP).

1-4. Void Formation in the HTL

The manuscript attributes void formation in the hole transport layer (HTL) to the use of low-boiling-point additives such as 4-tert-butylpyridine, which is reasonable. However, the observation that A82 (2-Phenylpyridine) does not induce void formation, while A15 (4-Phenylpyridine) does, requires further explanation. The authors should clarify whether they have investigated the thermodynamic properties of complexes formed between these additives and LiTFSI. Additionally, have they measured the glass transition temperature (T_g) of HTLs containing different additives? This data could provide insight into differences in void formation behavior.

Answer: Thank you for your pointing. To investigate the thermodynamic properties and to understand why 2-phenylpyridine does not induce void formation while 4-phenylpyridine does, we conducted differential scanning calorimetry (DSC). Through this measurement, we aimed to determine the glass transition temperature (T_g) of HTLs containing different additives. However, due to poor reproducibility, it was not possible to clearly identify the T_g . This may be attributed to residual solvents in Spiro-OMeTAD causing low reproducibility. Although we attempted solvent removal via pre-heating and/or vacuum treatment, reproducibility did not improve. Therefore, we would like to present these results only for the reviewers' reference.

Only for reviewers. Reproducibility check for DSC measurement of Spiro-OMeTAD containing

LiTFSI and 4-phenylpyridine. Three identical samples (sample 1, 2, and 3) were in the same condition and prepared for checking reproducibility. Spiro-OMeTAD solution was spin-coated onto the glass substrate and then peeled off to prepare the samples.

As an alternative aspect, density functional theory (DFT) calculations of the most stable configurations when the additives coordinate with lithium revealed significantly different geometries between 2-phenylpyridine and 4-phenylpyridine. Notably, 2-phenylpyridine formed a more compact coordination structure than 4-phenylpyridine. This difference in coordination may affect to thermodynamic properties, resulting in the lower volatility and absence of voids observed in Spiro-OMeTAD with 2-phenylpyridine under the 85°C test conditions. We noted this speculation in the manuscript. (see Comment 5 for more details)

Figure 4. DFT calculated chemical structure with **a**, 4-tert-butylpyridine + lithium **b**, 2-phenylpyridine + lithium, and viewed from different angles of **c**, 4-tert-butylpyridine + lithium

and **d**, 2-phenylpyridine + lithium.

The sentence below was added on page 19.

Furthermore, 2-phenylpyridine (A82) formed a significantly more compact coordination structure compared to the other additives, in which the lithium ion appeared to be tightly surrounded. This difference in coordination may be related to the lower volatility and absence of voids observed in Spiro-OMeTAD with 2-phenylpyridine under the 85°C test conditions.

Another possible mechanism for void formation can be attributed to the migration of the dopant from spiro-OMeTAD to the perovskite layer (see Reviewer 2, Comment 6).

1-5. Reactivity Between Perovskite and Pyridine Derivatives

The study reports that spin-coating pyridine derivatives directly onto perovskite films leads to degradation. However, in standard device fabrication, spin-coating HTL solutions containing 4-tert-butylpyridine does not appear to significantly impact device performance. The authors should clarify whether they observed the emergence of new XRD diffraction peaks in aged devices, similar to those found in direct pyridine spin-coating experiments. Furthermore, the claim that steric hindrance differences between 4-phenylpyridine and 3-phenylpyridine significantly impact coordination behavior is not well-supported. The authors should provide further discussion or computational evidence to substantiate this point.

Answer: Thank you for your valuable comments. We attempted to observe new XRD diffraction peaks in aged devices.

Supplementary Fig. S76 shows the sample structure used for XRD measurements, Figure 4a is the XRD patterns before aging, and Figure 4b is those after 100 hours of aging at 85 °C. The samples were prepared using the same recipe of device fabrication, incorporating 4-tert-butylpyridine, 4-phenylpyridine, 3-phenylpyridine, or 2-phenylpyridine.

Before aging (Figure 4a), the sample containing 4-phenylpyridine showed a small peak at 6.9°, which may be attributed to a PbI₂-additive complex.[ref] After 85 °C aging (Figure 4b), both the 4-tert-butylpyridine and 4-phenylpyridine samples exhibited a clear peak at 6.9°, suggesting the formation of a PbI₂-additive complex. In contrast, the samples containing 3-phenylpyridine and 2-phenylpyridine showed a peak at 12.7°, corresponding to PbI₂ generated during aging.

These results suggest that 4-tert-butylpyridine and 4-phenylpyridine may react with PbI₂ during aging, resulting in the formation of a PbI₂-additive complex. Although it is unclear how much this compound induces degradation, it may have some influence on the device durability.

The sentences were revised on page 17.

The XRD patterns (Fig. 4a,b) were analyzed to evaluate the reactivity between the additives and the perovskite layer comparing before and after 85°C tests. Samples were prepared same as device structure of FTO glass/SnO₂/perovskite (PVK)/Spiro-OMeTAD(+ additives)/Au

(Supplementary Fig. S76). Au layer was peeled out just before XRD measurement. Additives of 4-*tert*-butylpyridine (tBP), 4-phenylpyridine (A15), 3-phenylpyridine (A81), or 2-phenylpyridine (A82) were used for this experiment. For all the samples appeared peaks at 13.9°, 19.8° and 24.3°, which are associated with (100), (110), and (111) of perovskite crystal, respectively. Before 85°C test (Fig. 4a), the sample with 4-phenylpyridine (A15) showed a small peak at 6.9° that could be from PbI₂-additives complex.⁹¹ After 85 °C tests, samples with 4-*tert*-butylpyridine (tBP) and 4-phenylpyridine (A15) exhibit clear peak at 6.9° that could be from PbI₂-additives complex. These are possibly harmful by-products leading to the instability of PSCs. It seems that these by-products themselves are not the primary reason for poor thermal stability, but it can be the secondary reason. Interestingly, samples with 2-phenylpyridine (A82) and 3-phenylpyridine (A81) appear no peak at 6.9°, but small peaks were observed at 12.6 corresponding to PbI₂. These results indicate that 2-phenylpyridine and 3-phenylpyridine are low reactivity with the perovskite layer. This lack of reactivity is attributed to the steric hindrance introduced by the meta and ortho substitutions, which can prevent coordination with PbI₂, thereby contributing to their excellent thermal stability (Fig. 2b).

Supplementary Figure S76. Sample structure for XRD measurement. Each layer was deposited using the same recipe for device fabrication. Au layer was peeled out by using tape just before XRD measurement.

Figure 4. XRD patterns of devices with each additives. **a**, XRD patterns of samples with 4-tert-butylpyridine (tBP), 4-phenylpyridine (A15), 3-phenylpyridine (A81), and 2-phenylpyridine (A82). **b**, those of XRD patterns after 85 °C aging for 100 hours.

To investigate the effect of the steric hindrance differences on coordination behavior, we calculated the most stable configurations when the additives coordinate with lithium, which revealed different geometries. Figure 4e-l shows the DFT calculated chemical structure with 4-tert-butylpyridine (tBP), 4-phenylpyridine (A15), 3-phenylpyridine (A81), and 2-phenylpyridine (A82). All additives were found to form a tetrahedral arrangement as the most stable configuration, with point group S₄. 4-tert-Butylpyridine and 4-phenylpyridine were the similar geometries. Interestingly, 3-phenylpyridine exhibited a slightly more compact structure than 4-phenylpyridine due to geometric distortion induced by its molecular orientation. Furthermore, 2-phenylpyridine formed a significantly more compact coordination structure compared to the other additives, in which the lithium ion appeared to be tightly surrounded. This result was explained in the manuscript.

The sentences were revised on page 21.

The interaction between additives and lithium was analyzed using DFT calculations (Fig. 4e-l). Figures 4e-h illustrate the coordination of 4-tert-butylpyridine (tBP) with lithium, 4-phenylpyridine (A15) with lithium, 3-phenylpyridine (A81) with lithium 2-phenylpyridine (A82), respectively, and Figures 4i-l are corresponding structures viewed from different angles. All additives were found to form a tetrahedral arrangement as the most stable configuration, all with

point group S_4 . 4-tert-Butylpyridine (tBP) and 4-phenylpyridine (A15) were the similar geometries. Interestingly, 3-phenylpyridine exhibited a slightly more compact structure than 4-phenylpyridine (A15) due to geometric distortion induced by its molecular orientation. Furthermore, 2-phenylpyridine (A82) formed a significantly more compact coordination structure compared to the other additives, in which the lithium ion appeared to be tightly surrounded. This difference in coordination may be related to the lower volatility and absence of voids observed in Spiro-OMeTAD with 2-phenylpyridine under the 85°C test conditions. When three molecules of each additive were coordinated with lithium, an energetically favorable state of 2-phenylpyridine, 3-phenylpyridine, and 4-phenylpyridine was point group C_3 while tBP was D_3 , respectively. The desorption energy required to remove a coordinated molecule from lithium was calculated to be 0.667 eV for tBP, 0.659 eV for A15, 0.680 eV for A81, and 0.100 eV for A82, indicating that there is no relationship between thermal stability and ease of molecular desorption. Supplementary Figures S78a,b depict the interactions between the coordinated additive-lithium complexes and Spiro-OMeTAD. The influence of π -interactions between the methoxyphenylethylamine group in Spiro-OMeTAD and the pyridine rings of the additives was evaluated. Calculations show that a positive formation energy is required for interaction between tBP and A15 and Spiro-OMeTAD to happen because steric effects require a large energy to deform Spiro-OMeTAD to allow such interactions to happen. These results are consistent with the X-ray photoelectron spectroscopy, which shows no significant chemical shift (Supplementary Fig. S71). These results suggest that π -interactions between the additives and Spiro-OMeTAD do not affect the overall stability or performance of the system.

Figure 4. DFT calculated chemical structure with **e**, 4- tert-butylpyridine (tBP) and lithium. **f**, 4-Phenylpyridine (A15) and lithium. **g**, 3-Phenylpyridine (A81) and lithium. **h**, 2-Phenylpyridine (A82) and lithium. **i-l**, Corresponding structures viewed from different angles.

1-6. Typographical Errors

Minor typographical errors exist throughout the manuscript, such as "phyenylpyridine" instead of "phenylpyridine" in the Figure 3 captions. A thorough proofreading is recommended to improve readability and ensure accuracy.

Answer: Thank you for pointing out. We corrected typographical errors.

Reviewer #2 (Remarks to the Author):

The authors have done a substantial amount of work comparing the effects of different tBP substitutes on improving the thermal stability of doped Spiro-OMeTAD. This is a commendable effort. Although the author provides extensive thermal stability test data demonstrating the enhancement of device thermal stability with 3-phenylpyridine and 2-phenylpyridine, the explanation of the mechanism is quite lacking. More characterisation and analysis are needed to substantiate the role of 3-phenylpyridine and 2-phenylpyridine. Hope the following questions can be addressed in the revised manuscript.

Answer: Thank you so much for your valuable comments. We revised manuscript according to comments.

2-1. For the thermal stability, SnO₂ dispersed solution was used to fabricate SnO₂ carrier-transport-layer and CsFAPbI₃ was used to fabricate perovskite, whereas for the efficiency part, the carrier-transport-layer was changed to CBD SnO₂, and the perovskite was changed to RbCsFAPbI₃ with excess PbI₂ in precursor. It is necessary to provide the stability comparison data of highly efficient PSCs (dispersed-solution-SnO₂ vs CBD, CsFAPbI₃ vs RbCsFAPbI₃ with excess PbI₂).

Answer: Thank you for your pointing out. We provided stability comparison data of dispersed-solution-SnO₂ vs CBD and CsFAPbI₃ vs RbCsFAPbI₃. To provide high efficiency PSCs, we optimized the recipe for perovskite fabrication for each condition, which is explained at experimental part. These results indicate the way to approach both high efficiency and high stability.

In terms of dispersed-solution SnO₂ (SnO₂_DS) vs. CBD SnO₂ (SnO₂_CBD), the initial efficiency of SnO₂_CBD with 2-phenylpyridine was 20.1% and remained at 18.7% after 936 hours of testing at 85 °C. In contrast, SnO₂_DS with 2-phenylpyridine showed an initial efficiency of 14.9% and retained 16.2% after 936 hours at 85 °C. In both cases, 2-phenylpyridine provided higher stability compared with 4-tert-butylpyridine. From these results, SnO₂_CBD exhibited higher initial efficiency and may also offer excellent stability, note that, it is difficult to draw a

definitive conclusion based on this data alone.

As for CsFAPbI₃ vs RbCsFAPbI₃, rubidium doped perovskite (RbCsFAPbI₃) achieved higher efficiency (24.2%) than CsFAPbI₃ (20.1%), however, stability with RbCsFAPbI₃ showed significantly dropped after 87 hours of 85 °C test. From this result, It is possible that rubidium doped perovskite itself has problems with thermal durability. Correa-Baena et al. indicated that a dosage of 1% Rb was enough for causing segregation, which is a different kind of issue from those associated with additives¹⁰⁰. These results suggest that optimization of perovskite composition is also important to achieve high thermal stability.

Figure S71. Stability comparison of perovskite solar cells. (a) dispersed-solution-SnO₂ (SnO₂_DS) vs chemical bath deposition (SnO₂_CBD). (b) CsFAPbI₃ vs RbCsFAPbI₃.

The sentences were added to page 13.

We also provided stability comparison data of dispersed-solution-SnO₂ (SnO₂_DS) vs

chemical bath deposition (SnO_2 _CBD) as well as CsFAPbI_3 vs RbCsFAPbI_3 (Supplementary Fig. S71). To provide high efficiency PSCs, we optimized the recipe for perovskite fabrication for each condition, which is explained at experimental part. In terms of dispersed-solution SnO_2 (SnO_2 _DS) vs. CBD SnO_2 (SnO_2 _CBD), the initial efficiency of SnO_2 _CBD with 2-phenylpyridine was 20.1% and remained at 18.7% after 936 hours of testing at 85 °C. In contrast, SnO_2 _DS with 2-phenylpyridine showed an initial efficiency of 14.9% and retained 16.2% after 936 hours at 85 °C. In both cases, 2-phenylpyridine provided higher stability compared with 4-tert-butylpyridine. As for CsFAPbI_3 vs RbCsFAPbI_3 , rubidium doped perovskite (RbCsFAPbI_3) achieved higher efficiency (24.2%) than CsFAPbI_3 (20.1%), however, stability with RbCsFAPbI_3 showed significantly dropped after 87 hours of 85 °C test. From this result, It is possible that rubidium doped perovskite itself has problems with thermal durability. Correa-Baena et al. indicated that a dosage of 1% Rb was enough for causing segregation, which is a different kind of issue from those associated with additives¹⁰⁰. These results suggest that optimization of perovskite composition is also important to achieve high thermal stability.

2-2. It is necessary to provide the IPCE data and integrated current density of highly efficient PSCs shown in Fig.5c.

Answer: Thank you for your suggestions. We have provided the IPCE data and the integrated current density of the highly efficient PCSs in Fig. 5c. The integrated current density was 25.14 mA/cm² for 4-*tert*-butylpyridine, 25.33 mA/cm² for 4-phenylpyridine, and 25.39 mA/cm² for 2-phenylpyridine. The difference between J_{sc} obtained from the IV curves and the integrated current density was within 4%, which is acceptable considering recent publications in *Nature Communications*, *Nature*, etc. These results have been included in the supplementary information.

Figure S83. IPCE of perovskite solar cells with 4-*tert*-butylpyridine (tBP), 4-phenylpyridine (A15), and 2-phenylpyridine (A82).

2-3. Solely using DFT to confirm the interaction between additives, lithium and Spiro-OMeTAD is not sufficient. Fourier-transform infrared spectroscopy (FTIR) is needed to evidence the presence of interaction.

Answer: Thank you for your valuable comments. We measured FTIR spectra to investigate the molecular interactions. The hole-transport layer (Spiro-OMeTAD + LiTFSI + dopant) was examined by comparing 4-tert-butylpyridine, 4-phenylpyridine, and 2-phenylpyridine. Overall, the spectra of 4-phenylpyridine and 2-phenylpyridine were similar to that of 4-tert-butylpyridine. Peaks at 1061 cm^{-1} and 1349 cm^{-1} were observed, which correspond to the S=O stretching vibration of TFSI. Interestingly, these peaks were shifted to higher wavenumber in the cases of 4-phenylpyridine and 2-phenylpyridine. This shift could be attributed to the stronger coordination of 4-phenylpyridine and 2-phenylpyridine with lithium, which reduces TFSI coordination and thereby strengthens the S=O bond. These results could relate to the interaction between additives confirmed by DFT.

The sentences were revised on page 19.

We measured Fourier-transform infrared spectroscopy (FTIR) to investigate the molecular interactions (Supplementary Fig. S79). Peaks at 1061 cm^{-1} and 1349 cm^{-1} were observed, which correspond to the S=O stretching vibration of TFSI^{101,102}. Interestingly, these peaks were shifted to higher wavenumber in the cases of 4-phenylpyridine and 2-phenylpyridine. This shift could be attributed to the stronger coordination of 4-phenylpyridine and 2-phenylpyridine with lithium, which reduces TFSI coordination and thereby strengthens the S=O bond.

Figure S79. FTIR spectra of Spiro-OMeTAD with LiTFSI and different dopants: (a) full spectrum and expanded views of (b) $1055\text{--}1066\text{ cm}^{-1}$ and (c) $1345\text{--}1355\text{ cm}^{-1}$.

2-4. Traditional Spiro-OMeTAD HTL suffers crystallization after thermal aging which worsens the electrical properties of HTL. Therefore, it is necessary to provide AFM and microscope photographs of the HTLs before and after thermal aging.

Answer: Thank you for your valuable comments. We performed AFM and optical microscopy measurements of the HTLs before and after thermal aging (85 °C for 100 hours). Sample structure was FTO glass/SnO₂/Perovskite/Spiro-OMeTAD/Au. Spiro-OMeTAD layer contains LiTFSI and each additive same as device fabrication. Au layer was peeled out by tape just before measurement. After the 85 °C test, the HTL with 4-tert-butylpyridine exhibited pin-hole formation (Supplementary Fig. S74d) and cracks originating from crystalline-like regions (Figure S74e). These results are consistent with previous reports⁹⁷. The depths of the pin-holes and cracks in the crystalline-like regions were in the range of 50–150 nm. Such defects are expected to deteriorate the electrical properties of the HTLs as well as the interfaces between HTL/Au and Perovskite/HTL. In contrast, the HTLs with 4-phenylpyridine and 2-phenylpyridine showed no significant morphological changes after the 85 °C test. These observations suggest that 4-phenylpyridine and 2-phenylpyridine effectively suppress pin-hole and crack formation in crystalline-like regions, which account for their higher thermal stability.

The sentences were revised on page 17.

Furthermore, we performed optical microscopy and AFM measurements of the HTLs before and after thermal aging (85 °C for 100 hours, Supplementary Fig. S73, S74). After thermal aging, the HTL with 4-tert-butylpyridine exhibited pin-hole formation and cracks originating from crystalline-like regions. These results are consistent with previous reports⁹⁶. The depths of the pin-holes and cracks in the crystalline-like regions were in the range of 50–150 nm. Such defects are expected to deteriorate the electrical properties of the HTLs as well as the interfaces between HTL/Au and Perovskite/HTL. In contrast, the HTLs with 4-phenylpyridine and 2-phenylpyridine showed no significant morphological changes after the 85 °C test. These observations suggest that 4-phenylpyridine and 2-phenylpyridine effectively suppress pin-hole and crack formation in crystalline-like regions, which account for their higher thermal stability.

Figure 73. Optical microscope images of the HTLs before and after thermal test. Additives with (a) 4-tert-butylpyridine, (b) 4-phenylpyridine, (c) 2-phenylpyridine and (d-f) corresponding additives after 85 °C test for 100 hours. Sample structure was FTO glass/SnO₂/Perovskite/Spiro-OMeTAD/Au. Spiro-OMeTAD layer contains LiTFSI and each additives same as device fabrication. Au layer was peeled out by tape just before measurement. Scale bar corresponds to 30 μm.

Figure 74. AFM images of the HTLs before and after thermal aging. Additives with (a) 4-tert-butylpyridine, (b) 4-phenylpyridine, (c) 2-phenylpyridine, and (d-g) corresponding additives after 85 °C test for 100 hours. Sample structure was FTO glass/SnO₂/Perovskite/Spiro-OMeTAD/Au. Spiro-OMeTAD layer contains LiTFSI and each additives same as device fabrication. Au layer was peeled out by tape just before measurement. Scale bar corresponds to 5 μm.

2-5. Additives could influence the oxidation of hole-transport layer. In this work, does the best two additives influence the oxidation/doping progress of hole-transport layer? Accelerate or postpone it? It is necessary to provide UV-vis absorption data to analyse the HTL before and after oxidation. UPS/KPFM is also necessary to investigate the associated working function/energy level changes.

Answer: Thank you for your valuable comments.

UV-vis absorption spectroscopy was conducted to assess the potential influence of the additives on the oxidation of the hole-transport layer. The spiro-OMeTAD films were prepared with LiTFSI, 4-tert-butylpyridine, 4-phenylpyridine, and 2-phenylpyridine, following the standard device fabrication protocol. The UV-vis spectra exhibited no appreciable differences among the samples, and no characteristic oxidation-related absorption peaks were detected. These findings indicate that neither 4-phenylpyridine nor 2-phenylpyridine exerts a measurable effect on the oxidation process.

Figure S86. (a) UV-vis spectra and (b) bandgap of Spiro-OMeTAD with additives. Bandgap was 3.02 eV.

PYS/KPFM is also measured to investigate the associated working function/energy level changes. Photoelectron yield spectroscopy (PYS) was used to measure the valence band instead of UPS. From these data, energy diagrams were calculated as shown supplementary Figure S89. Valence band edges (E_v) of Spiro-OMeTAD with 4-tert-butylpyridine was -5.58 eV. Interestingly,

E_v of Spiro-OMeTAD with 4-phenylpyridine, 3-phenylpyridine, and 2-phenylpyridine were -5.76 eV, -5.76 eV, and -5.83 eV, which is closer to perovskite (-5.89 eV) than 4-tert-butylpyridine. This could be the reason for effective charge transfer and improved photoconversion efficiency.

The sentences were revised on page 23.

UV-vis absorption spectroscopy was performed to evaluate the potential influence of the additives on the oxidation of the hole-transport layer (Supplementary Figure S86). The UV-vis spectra showed no appreciable differences among the samples, suggesting that neither 4-phenylpyridine nor 2-phenylpyridine has a measurable effect on the oxidation process. Furthermore, energy level diagrams of Spiro-OMeTAD with additives were constructed based on photoelectron yield spectroscopy and Kelvin probe force microscopy (KPFM) (Supplementary Fig. S87-89). The valence band edge (E_v) of Spiro-OMeTAD with 4-tert-butylpyridine was -5.58 eV. In contrast, the E_v values of Spiro-OMeTAD with 4-phenylpyridine, 3-phenylpyridine, and 2-phenylpyridine were -5.76 eV, -5.76 eV, and -5.83 eV, respectively, which are closer to that of perovskite (-5.89 eV) than the value obtained with 4-tert-butylpyridine. This alignment is likely to facilitate more efficient charge transfer, thereby contributing to improved photoconversion efficiency.

Figure S87. Photoelectron yield spectroscopy of (a) perovskite, (b) perovskite with OAI passivation, (c) spiro-OMeTAD with LiTFSI and 4-tert-butylpyridine, (d) spiro-OMeTAD with LiTFSI and 4-phenylpyridine, (e) spiro-OMeTAD with LiTFSI and 3-phenylpyridine, (f) spiro-OMeTAD with LiTFSI and 2-phenylpyridine.

Figure S88. KPFM mapping of (a) perovskite, (b) perovskite with OAI passivation, (c) spiro-OMeTAD with LiTFSI and 4-tert-butylpyridine, (d) spiro-OMeTAD with LiTFSI and 4-phenylpyridine, (e) spiro-OMeTAD with LiTFSI and 3-phenylpyridine, (f) spiro-OMeTAD with LiTFSI and 2-phenylpyridine. Scale bar corresponds to 5 μm.

Figure S89. Energy diagram of perovskite and spiro-OMeTAD with additives. Energy diagrams were calculated with photoelectron yield spectroscopy, KPFM, and bandgap from UVVIS spectroscopy. Bandgap of perovskite and phenylpyridine was associated with 1.56 eV and 3.02 eV, respectively. Spiro-OMeTAD include LiTFSI and each additive same as recipe with device fabrication.

2-6. Lithium salt was observed to accumulate at interfaces (Perovskite solar cells based on Spiro-OMeTAD stabilized with an alkylthiol additive *Nature Photonics*, 2023, 17, 96-105). Are the additives proposed in this work helpful to obtain homogenous dopant distribution? Also, it is important to locate the position of additives. Is it homogeneously distributed throughout the HTL? Therefore, time-of-flight secondary-ion mass spectrometry (TOF-SIMS) or other characterization methods which can clearly demonstrate the out-of-plane component distribution is important to help readers understand the benefits of new the additives proposed in this work.

Answer: Thank you for your valuable comments. We investigated the effect of additives on elemental distribution using ToF-SIMS (Supplementary Fig. S80). The measured samples were prepared with the configuration FTO/SnO₂/Perovskite/Spiro-OMeTAD (with additives)/Au, following the same procedure as device fabrication. The Au layer was peeled off immediately before ToF-SIMS measurement. We compared 4-*tert*-butylpyridine and 2-phenylpyridine before and after thermal aging at 85 °C for 100 hours. Lithium migrated toward the ETL side and slightly accumulated at the surface, which is consistent with a previous report¹⁰³. The lithium distribution was not significantly affected by the choice of dopant, nor by the thermal test. From these results, 2-phenylpyridine does not appear to promote a more homogeneous lithium distribution.

For the fresh device with 4-*tert*-butylpyridine, the additive was mainly distributed in the Spiro-OMeTAD layer (Supplementary Fig. S80a). After the 85 °C test, however, 4-*tert*-butylpyridine had migrated substantially into the perovskite layer (Supplementary Fig. S80b). This migration is likely driven by its reaction with the perovskite, which is consistent with the by-product confirmed by XRD (Figure 4b). In contrast, 2-phenylpyridine was mainly localized in the Spiro-OMeTAD layer, and its distribution showed no significant change after the 85 °C test (Supplementary Fig. S80c,d). This stability may contribute to its beneficial effect on thermal stability, probably because 2-phenylpyridine does not react with the perovskite and therefore does not diffuse into the perovskite layer.

Furthermore, the ToF-SIMS results may help explain the absence of voids in the 2-phenylpyridine-containing Spiro-OMeTAD layer after the 85 °C test. In the case of 4-*tert*-butylpyridine, its significant migration from Spiro-OMeTAD into the perovskite layer may have contributed to void formation within Spiro-OMeTAD (Supplementary Fig. S81a,b). By contrast, the void-free morphology observed with 2-phenylpyridine can be attributed to the stability of its

distribution (Supplementary Fig. S81c,d).

Although it remains difficult to determine the precise mechanism of void formation from these results alone, they may provide valuable design guidelines for void-free HTMs.

The sentences were revised on page 21.

For further understanding of mechanism of the void-formation, we investigated the effect of additives on elemental distribution using ToF-SIMS (Supplementary Fig. S80). The measured samples were prepared with the configuration FTO/SnO₂/Perovskite/Spiro-OMeTAD (with additives)/Au, following the same procedure as device fabrication. The Au layer was peeled off immediately before ToF-SIMS measurement. We compared 4-tert-butylpyridine and 2-phenylpyridine before and after thermal aging at 85 °C for 100 hours. Lithium migrated toward the ETL side and slightly accumulated at the surface, which is consistent with a previous report¹⁰³. The lithium distribution was not significantly affected by the choice of dopant, nor by the thermal test. From these results, 2-phenylpyridine does not appear to promote a more homogeneous lithium distribution.

For the fresh device with 4-tert-butylpyridine, the additive was mainly distributed in the Spiro-OMeTAD layer (Supplementary Fig. S80a). After the 85 °C test, however, 4-tert-butylpyridine had migrated substantially into the perovskite layer (Supplementary Fig. S80b). This migration is likely driven by its reaction with the perovskite, which is consistent with the by-product confirmed by XRD (Figure 4b). In contrast, 2-phenylpyridine was mainly localized in the Spiro-OMeTAD layer, and its distribution showed no significant change after the 85 °C test (Supplementary Fig. S80c,d). This stability may contribute to its beneficial effect on thermal stability, probably because 2-phenylpyridine does not react with the perovskite and therefore does not diffuse into the perovskite layer. These ToF-SIMS results may explain the absence of voids in the 2-phenylpyridine-containing Spiro-OMeTAD layer after the 85 °C test. In the case of 4-tert-butylpyridine, its significant migration from Spiro-OMeTAD into the perovskite layer may have contributed to void formation within Spiro-OMeTAD (Supplementary Fig. S81a,b). By contrast, the void-free morphology observed with 2-phenylpyridine can be attributed to the stability of its distribution (Supplementary Fig. S81c,d).

Figure S80. ToF-SIMS depth profiles of (a) a fresh device with 4-tert-butylpyridine, (b) the same device after the 85 °C test, (c) a fresh device with 2-phenylpyridine, and (d) the same device after the 85 °C test.

Figure S81. Schematics images of void formation and void-free by additives. (a) a fresh device

with 4-tert-butylpyridine, (b) the same device after the 85 °C test, (c) a fresh device with 2-phenylpyridine, and (d) the same device after the 85 °C test.

2-7. The poor thermal stability of doped Spiro-OMeTAD is not solely attributed to the additives (such as Li, tBP); it may also be related to intrinsic factors of Spiro-OMeTAD itself. For instance, its low T_g makes it susceptible to molecular segmental motion, and non-crystalline Spiro-OMeTAD can undergo local recrystallization at high temperatures. Additionally, the diffusion of I⁻ from the perovskite into HTL can also lead to the failure of Spiro-OMeTAD. Given this, why does the substitution of tBP with 3-phenylpyridine and 2-phenylpyridine result in a significant improvement in thermal stability? Please provide more evidence to determine whether 3-phenylpyridine and 2-phenylpyridine influence the factors mentioned above.

Answer: Thank you for insightful comments. To investigate the thermodynamic properties and to elucidate why 2-phenylpyridine does not induce void formation whereas 4-phenylpyridine does, differential scanning calorimetry (DSC) was performed. The objective was to determine the glass transition temperature (T_g) of HTLs incorporating different additives. However, the measurements exhibited poor reproducibility, precluding a clear identification of T_g. This inconsistency may be attributed to residual solvents in Spiro-OMeTAD. Attempts to improve reproducibility through pre-heating and/or vacuum treatment were unsuccessful. Accordingly, these results are provided here solely for the reviewers' reference. (see Reviewer 1, comment 4)

To investigate local recrystallization at high temperatures, we performed optical microscopy and AFM measurements of the HTLs before and after thermal aging (85 °C for 100 hours, Supplementary Figure S73, S74). (see Reviewer 2, comments 4) After thermal aging, the HTL with 4-tert-butylpyridine exhibited pin-hole formation and cracks originating from crystalline-like regions. These results are consistent with previous reports⁹⁶. The depths of the pin-holes and cracks in the crystalline-like regions were in the range of 50–150 nm. Such defects are expected to deteriorate the electrical properties of the HTLs as well as the interfaces between HTL/Au and Perovskite/HTL. In contrast, the HTLs with 4-phenylpyridine and 2-phenylpyridine showed no significant morphological changes after the 85 °C test. These observations suggest that 4-phenylpyridine and 2-phenylpyridine effectively suppress pin-hole and crack formation in crystalline-like regions, which account for their higher thermal stability.

As for the diffusion of I⁻, we conducted ToF-SIMS for devices before and after 85°C test

comparing 4-tert-butylpyridine and 2-phenylpyridine (Supplementary Fig. S80). I⁻ is mainly distributed in the perovskite layer and there was no significant changes in distribution after the 85 °C test regardless of the dopants. From this result, 2-phenylpyridine does not influence of the I⁻ distribution.

From these results, additive migration (see Comment 6) is most likely the underlying mechanism by which 2-phenylpyridine enhances thermal stability. In the case of 4-tert-butylpyridine, substantial migration from the Spiro-OMeTAD layer into the perovskite may have contributed to void formation within Spiro-OMeTAD (Supplementary Figure S81a,b). By contrast, the void-free morphology observed with 2-phenylpyridine can be attributed to the stability of its spatial distribution (Supplementary Figure S81c,d).

2-8. What are the boiling points of 3-phenylpyridine, 2-phenylpyridine, and tBP? Could the differences in the results shown in Figure 3 be due to the different boiling points of the three materials?

Answer: Thank you for your valuable comments. The boiling points are 274 °C for 3-phenylpyridine, 270 °C for 2-phenylpyridine, and 197 °C for 4-tert-butylpyridine, respectively. These values are consistent with the trends observed in Figure 3. 4-tert-butylpyridine is known to be highly volatile, which may lead to void formation after thermal stress. In contrast, 2-phenylpyridine and 3-phenylpyridine exhibit lower volatility, which can contribute to enhanced thermal stability. This reduced volatility, together with the stable additive distribution described in Comment 6, likely accounts for the superior thermal stability of 2-phenylpyridine and 3-phenylpyridine.

The sentences were revised on page 17.

The boiling points are 274 °C for 3-phenylpyridine, 270 °C for 2-phenylpyridine, and 197 °C for 4-tert-butylpyridine, respectively. These values are consistent with the trends observed in Figure 3.

2-9. Please provide the AFM images of Spiro-OMeTAD doped with the three additives, Li+3-phenylpyridine, Li+2-phenylpyridine, and Li+tBP, before and after heating, in order to observe the behavior of Li+ before and after heating.

Answer: Thank you for valuable comments. We performed AFM measurements of the HTLs before and after thermal aging (85 °C for 100 hours). Sample structure was FTO glass/SnO₂/Perovskite/Spiro-OMeTAD/Au. Spiro-OMeTAD layer contains LiTFSI and each additive same as device fabrication. (Please see Comment 4)

2-10. For stability, the thickness of Spiro-OMeTAD (120 mg/ml) has been increased in this manuscript. However, thicker Spiro-OMeTAD tends to increase the resistance of the HTL layer. Any trade-off benefit here?

Answer: Thank you for your comment. We examined whether the concentration of Spiro-OMeTAD affects device performance by comparing 30 mg/ml and 120 mg/ml solutions. For series resistance (R_s), the average R_s of the 30 mg/ml devices was $5.4 \Omega \text{ cm}^2$, slightly lower than that of the 120 mg/ml devices ($5.8 \Omega \text{ cm}^2$), which may be attributed to the thinner Spiro-OMeTAD layer. This negligible difference in resistance can be explained by the excellent resistivity of 4-phenylpyridine (see Reviewer 3, Comment 3). In contrast, the average shunt resistance (R_{sh}) of the 30 mg/ml devices was $7,854 \Omega \text{ cm}^2$, lower than that of the 120 mg/ml devices ($10,634 \Omega \text{ cm}^2$), resulting in reduced FF and PCE. This reduction may result from the thinner Spiro-OMeTAD layer, which could induce pinholes and reduce reproducibility. Regarding the champion devices, no significant differences were observed in photovoltaic properties. Overall, these results indicate that a higher Spiro-OMeTAD concentration (120 mg/ml) provides improved reproducibility without substantially increasing resistance.

The sentences were revised on page 23.

We also examined whether the concentration of Spiro-OMeTAD influences device performance by comparing 30 mg/ml and 120 mg/ml solutions (Supplementary Fig. S84, S85). The results indicate that the higher concentration (120 mg/ml) improves reproducibility without significantly increasing resistance.

Figure S84. Statics of photovoltaic performance of perovskite solar cells comparing 30 mg/mL and 120 mg/mL of 4-phenylpyridine in Spiro-OMeTAD solution for (a) J_{sc} , (b) V_{oc} , (c), FF, (d) PCE, (e) R_s , and (f) R_{sh} . Total number of samples was 16.

Figure S85. IV-curves of champion solar cells comparing 30 mg/mL and 120 mg/mL of Spiro-OMeTAD solution containing 4-phenylpyridine and LiTFSI.

Reviewer #3 (Remarks to the Author):

In this manuscript, the authors consider tBP as the key factor for the instability of Spiro-OMeTAD-based devices at high temperatures. They employed a large number of heteroaryl derivatives to verify which materials could improve thermal stability. Finally, the authors proposed that three types of derivatives have a positive effect on thermal stability, and based on this, devices were fabricated, which showed good stability in outdoor stability tests. Overall, the authors are thanked for their extensive work and experiments in screening suitable derivatives, and for providing reliable efficiency aging data instead of the normalized data in most literature. The authors also provided some mechanistic explanations to better understand the impact of such additives on the stability of Spiro-OMeTAD. However, despite the numerous experiments and tests conducted by the authors, the research results are not persuasive enough, and the mechanistic explanations lack new ideas, failing to arouse greater interest among readers. In addition, there are some logical issues in this research. Therefore, I do not recommend further consideration of this manuscript for publication in Nature Communications. The specific comments are as follows:
Answer: Thank you so much for your valuable comments. We revised manuscript according to comments.

3-1. The authors believe that the thermal instability of tBP stems from its volatilization at high temperatures, which causes pore problems in Spiro-OMeTAD, thereby leading to the thermal instability of the devices. However, for Spiro-OMeTAD-based devices, the interaction between tBP and Li salts cannot be ignored. The core function of tBP doping is to promote the dissolution of lithium salts and prevent the agglomeration of lithium salts. Nevertheless, in this study, the authors used a large number of additives without considering the dissolution effect of the materials on lithium salts. This may be a key factor affecting device performance and thermal stability, rather than the effect brought about by the structure of the derivatives themselves. The authors did not rule out this key factor in their own research.

Answer: Thank you for your insightful comments. We examined the dissolution effect of the additives on lithium salts. Lithium salts were dissolved in Spiro-OMeTAD solutions containing each additive. Supplementary Figure S75 presents the relationship between thermal stability (PCE after 1000 h) and the solubility of lithium salts. tBP and 3-phenylpyridine (A81) dissolved 62.5 mg of lithium salts, whereas 4-phenylpyridine (A15) and 2-phenylpyridine (A82) dissolved 55.6 mg and 11.4 mg, respectively. These results show no clear correlation between thermal stability and lithium salt solubility, suggesting that other mechanisms contribute to thermal stability.

The sentences were revised on page 18.

We also examined the dissolution effect of the additives on lithium salts. Since the additives promote the dissolution of lithium salts and prevent the agglomeration of lithium salts, this may be a key factor affecting device performance and thermal stability. Lithium salts were dissolved in Spiro-OMeTAD solutions containing each additive. Supplementary Fig. S75 presents the relationship between thermal stability (PCE after 1000 h) and the solubility of lithium salts. tBP and 3-phenylpyridine (A81) dissolved 62.5 mg/mL of lithium salts, whereas 4-phenylpyridine (A15) and 2-phenylpyridine (A82) dissolved 55.6 mg/mL and 11.4 mg/mL, respectively. These results show no clear correlation between thermal stability and lithium salt solubility, suggesting that other mechanisms contribute to thermal stability.

Supplementary Figure S75. PCE after 1000 hours vs solubility of LiTFSI with each additive. Lithium salts were dissolved in Spiro-OMeTAD solutions containing each additive.

To reveal the mechanism for thermal stability with novel additives, we investigated the effect of additives on elemental distribution using ToF-SIMS (Supplementary Fig. S80). The measured samples were prepared with the configuration FTO/SnO₂/Perovskite/Spiro-OMeTAD (with additives)/Au, following the same procedure as device fabrication. The Au layer was peeled off immediately before ToF-SIMS measurement. We compared 4-tert-butylpyridine and 2-phenylpyridine before and after thermal aging at 85 °C for 100 hours. Lithium migrated toward the ETL side and slightly accumulated at the surface, which is consistent with a previous report¹⁰³. The lithium distribution was not significantly affected by the choice of dopant, nor by the thermal test. From these results, 2-phenylpyridine does not appear to promote a more homogeneous lithium distribution.

For the fresh device with 4-tert-butylpyridine, the additive was mainly distributed in the Spiro-OMeTAD layer (Supplementary Fig. S80a). After the 85 °C test, however, 4-tert-butylpyridine had migrated substantially into the perovskite layer (Supplementary Fig. S80b). This migration is likely driven by its reaction with the perovskite, which is consistent with the by-product confirmed by XRD (Figure 4b). In contrast, 2-phenylpyridine was mainly localized in the Spiro-OMeTAD layer, and its distribution showed no significant change after the 85 °C test (Supplementary Fig. S80c,d). This stability may contribute to its beneficial effect on thermal stability, probably because

2-phenylpyridine does not react with the perovskite and therefore does not diffuse into the perovskite layer.

Furthermore, the ToF-SIMS results may help explain the absence of voids in the 2-phenylpyridine-containing Spiro-OMeTAD layer after the 85 °C test. In the case of 4-tert-butylpyridine, its significant migration from Spiro-OMeTAD into the perovskite layer may have contributed to void formation within Spiro-OMeTAD (Supplementary Figure S81a,b). By contrast, the void-free morphology observed with 2-phenylpyridine can be attributed to the stability of its distribution (Supplementary Figure S81c,d).

Although it remains difficult to determine the precise mechanism of void formation from these results alone, they may provide valuable design guidelines for void-free HTMs.

The sentences were revised on page 23.

We investigated the effect of additives on elemental distribution using ToF-SIMS. The measured samples were prepared with the configuration FTO/SnO₂/Perovskite/Spiro-OMeTAD (with additives)/Au, following the same procedure as device fabrication. The Au layer was peeled off immediately before ToF-SIMS measurement. We compared 4-tert-butylpyridine and 2-phenylpyridine before and after thermal aging at 85 °C for 100 hours. Lithium migrated toward the ETL side and slightly accumulated at the surface, which is consistent with a previous report¹⁰³. The lithium distribution was not significantly affected by the choice of dopant, nor by the thermal test. From these results, 2-phenylpyridine does not appear to promote a more homogeneous lithium distribution.

For the fresh device with 4-tert-butylpyridine, the additive was mainly distributed in the Spiro-OMeTAD layer. After the 85 °C test, however, 4-tert-butylpyridine had migrated substantially into the perovskite layer. This migration is likely driven by its reaction with the perovskite, which is consistent with the by-product confirmed by XRD (Figure 4b). In contrast, 2-phenylpyridine was mainly localized in the Spiro-OMeTAD layer, and its distribution showed no significant change after the 85 °C test. This stability may contribute to its beneficial effect on thermal stability, probably because 2-phenylpyridine does not react with the perovskite and therefore does not diffuse into the perovskite layer. These ToF-SIMS results may explain the absence of voids in the

2-phenylpyridine-containing Spiro-OMeTAD layer after the 85 °C test. In the case of 4-tert-butylpyridine, its significant migration from Spiro-OMeTAD into the perovskite layer may have contributed to void formation within Spiro-OMeTAD (Supplementary Figure S80a,b). By contrast, the void-free morphology observed with 2-phenylpyridine can be attributed to the stability of its distribution (Supplementary Figure S80c,d).

Figure S80. ToF-SIMS depth profiles of (a) a fresh device with 4-tert-butylpyridine, (b) the same device after the 85 °C test, (c) a fresh device with 2-phenylpyridine, and (d) the same device after the 85 °C test.

Figure S81. Schematics images of void formation and void-free by additives. (a) a fresh device with 4-tert-butylpyridine, (b) the same device after the 85 °C test, (c) a fresh device with 2-phenylpyridine, and (d) the same device after the 85 °C test.

3-2. Although the author provided a lot of device aging data for different additives in the supporting information, the device performance indicated by these data is really too low. Five years ago, it was very easy for devices using TBP to achieve an efficiency of over 20%, and it was not difficult to reach an efficiency of over 23% after passivation. However, the author only obtained an efficiency of around 15%, which indicates that there are significant problems with the perovskite thin film itself. The initial efficiency of the additives A80 and A81 proposed by the author is only above 14%, and using such data to measure the thermal stability of the devices lacks sufficient persuasiveness. In addition, after the modification of the electron transport layer (ETL) and passivation, the efficiency can be increased to 25%, but the final efficiency of the device modified with TBP is only 20.5%, which is hardly convincing.

Answer: Thank you for valuable comments. We tried to improve the efficiency of perovskite solar cells by optimization of fabrication recipe. To improve the photovoltaic performance of the perovskite solar cells with tBP, we optimized device perovskite recipe by increasing MACl concentration from 10% to 20%. By this optimization, devices with tBP achieved 0.771 of FF and 22.4% of PCE. These results were added to the manuscript.

Fig. 5 a $I-V$ scans of the perovskite solar cells with 4-tert-butylpyridine (Li+tBP). Note that MACl concentration increased from 10% to 20% to improve efficiency for devices with tBP.

Moreover, we optimized the efficiency of the 2-phenylpyridine (A82) by increasing the MACl concentration from 10% to 20%. By this optimization, devices with 2-phenylpyridine obtained

20.1% as an initial PCE and achieved improved thermal stability. Also, we conducted the 85 °C test, showing excellent thermal stability with improved PCE. These results were included in the supplementary information. Although this work focuses on the thermal stability of the perovskite solar cells rather than their photoconversion efficiency, we think that these improved photoconversion efficiency and high thermal stability result are sufficient for stability study.

Figure S71. Stability comparison of perovskite solar cells. (a) dispersed-solution-SnO₂ (SnO₂_DS) vs chemical bath deposition (SnO₂_CBD). (b) CsFAPbI₃ vs RbCsFAPbI₃.

3-3. It is insufficient to merely use TRPL to characterize the performance of the Spiro-OMeTAD thin films after the addition of different additives. If the modified Spiro-OMeTAD has sufficient performance advantages, the author should provide more characteristics of the optimized Spiro-OMeTAD, including the energy band, electrical conductivity, and so on.

Answer: Thank you for valuable comments. We investigated energy band and electrical conductivity to provide more characteristics. Also, UV/vis and FTIR measurement were also conducted. (see reviewer 2, Comment 3 and 5)

PYS/KPFM is measured to investigate the associated working function/energy level changes. Photoelectron yield spectroscopy (PYS) was used to measure the valence band instead of UPS. From these data, energy diagrams were calculated as shown supplementary Figure S89. Valence band edges (E_v) of Spiro-OMeTAD with 4-tert-butylpyridine was -5.58 eV. Interestingly, E_v of Spiro-OMeTAD with 4-phenylpyridine, 3-phenylpyridine, and 2-phenylpyridine were -5.76 eV, -5.76 eV, and -5.83 eV, which is closer to perovskite (-5.89 eV) than 4-tert-butylpyridine. This could be the reason for effective charge transfer and improved photoconversion efficiency.

The sentences were revised on page 26.

Furthermore, energy level diagrams of Spiro-OMeTAD with additives were constructed based on photoelectron yield spectroscopy and Kelvin probe force microscopy (KPFM) (Supplementary Fig. S87-89). The valence band edge (E_v) of Spiro-OMeTAD with 4-tert-butylpyridine was -5.58 eV. In contrast, the E_v values of Spiro-OMeTAD with 4-phenylpyridine, 3-phenylpyridine, and 2-phenylpyridine were -5.76 eV, -5.76 eV, and -5.83 eV, respectively, which are closer to that of perovskite (-5.89 eV) than the value obtained with 4-tert-butylpyridine. This alignment is likely to facilitate more efficient charge transfer, thereby contributing to improved photoconversion efficiency.

Figure S87. Photoelectron yield spectroscopy of (a) perovskite, (b) perovskite with OAI passivation, (c) spiro-OMeTAD with LiTFSI and 4-tert-butylpyridine, (d) spiro-OMeTAD with LiTFSI and 4-phenylpyridine, (e) spiro-OMeTAD with LiTFSI and 3-phenylpyridine, (f) spiro-OMeTAD with LiTFSI and 2-phenylpyridine.

Figure S88. KPFM mapping of (a) perovskite, (b) perovskite with OAI passivation, (c) spiro-

OMeTAD with LiTFSI and 4-*tert*-butylpyridine, (d) spiro-OMeTAD with LiTFSI and 4-phenylpyridine, (e) spiro-OMeTAD with LiTFSI and 3-phenylpyridine, (f) spiro-OMeTAD with LiTFSI and 2-phenylpyridine. Scale bar corresponds to 5 μm.

Figure S89. Energy diagram of perovskite and spiro-OMeTAD with additives. Energy diagrams were calculated with photoelectron yield spectroscopy, KPFM, and bandgap from UVVIS spectroscopy. Bandgap of perovskite and phenylpyridine was associated with 1.56 eV and 3.02 eV, respectively. Spiro-OMeTAD include LiTFSI and each additive same as recipe with device fabrication.

In terms of conductivity, the resistivity of Spiro-OMeTAD layers containing LiTFSI and each additive was measured using the van der Pauw method (Supplementary Fig. S90). The Spiro-OMeTAD layers were deposited on glass substrates by spin-coating. Spiro-OMeTAD solutions were prepared using the same procedure as in the device fabrication. The resistivity of the 4-*tert*-butylpyridine, 4-phenylpyridine, 3-phenylpyridine, and 2-phenylpyridine were 19,100 Ω cm, 8,540 Ω cm, 6,050 Ω cm, and 2,680 Ω cm, respectively. From these results, 2,3,4-phenylpyridine

have an advantage in the conductivity compared to 4-tert-butylpyridine, resulting in improved photovoltaic performance.

The sentences were revised on page 26.

The resistivity of Spiro-OMeTAD layers containing LiTFSI and each additive were 19,100 Ω cm for 4-tert-butylpyridine, 8,540 Ω cm for 4-phenylpyridine, 6,050 Ω cm for 3-phenylpyridine, and 2,680 Ω cm for 2-phenylpyridine, respectively (Supplementary Figure S90). These results suggest that 2,3,4-phenylpyridine have an advantage in the resistivity compared to 4-tert-butylpyridine, resulting in improved FF and PCE.

Figure S90. The resistivity of Spiro-OMeTAD layers containing LiTFSI and each additive was measured using the van der Pauw method. The Spiro-OMeTAD layers were deposited on glass substrates by spin-coating. Spiro-OMeTAD solutions were prepared using the same procedure as in the device fabrication.

3-4. The author spent a great deal of space in the previous part of the manuscript on using additives to address the thermal instability of perovskite devices. However, in the subsequent characterization of the optimized devices, the author tested the operational stability under actual outdoor ambient. Although this outdoor stability is crucial for perovskite devices, I fail to find sufficient logical coherence between the two parts. According to the temperature diagram provided by the author in the SI, it seems that the stability at high temperatures is not covered. Instead, the author emphasizes the performance under diurnal cycles. Therefore, I believe there are significant logical issues in this section. The author should focus on and provide the thermal stability performance when the efficiency is optimized to over 25%.

Answer: Thank you for valuable comments. High efficiency and durability often involve a trade-off relationship. Given this situation, we optimized the device to achieve both high efficiency and durability as much as possible. As a result, a device incorporating 2-phenylpyridine achieved an initial PCE of 20.1% and retained 18.7% after 936 hours at 85 °C. These results are included in the manuscript.

In contrast, rubidium-doped perovskite (RbCsFAPbI₃) exhibited higher efficiency (24.2%) compared to CsFAPbI₃ (20.1%). However, the stability of RbCsFAPbI₃ dropped significantly after only 87 hours at 85 °C, suggesting that rubidium-doped perovskites themselves may have issues with thermal durability. Correa-Baena et al. reported that a dosage of only 1% Rb was sufficient to induce segregation, which represents a different type of problem compared with those associated with additives (https://www.sciencedirect.com/science/article/pii/S266693582100104X?utm_source=chatgpt.com). These results highlight that optimization of perovskite composition is also crucial for achieving high thermal stability. The corresponding data are provided in the Supplementary Information.

Although the main focus of this work is the thermal stability of perovskite solar cells rather than their power conversion efficiency, we think that the observed improvements in both efficiency and thermal stability are sufficiently relevant to the stability study.

Figure S71. Stability comparison of perovskite solar cells. (a) dispersed-solution-SnO₂ (SnO₂_DS) vs chemical bath deposition (SnO₂_CBD). (b) CsFAPbI₃ vs RbCsFAPbI₃.

The sentences were added to supplementary information.

We also provided stability comparison data of dispersed-solution-SnO₂ (SnO₂_DS) vs chemical bath deposition (SnO₂_CBD) as well as CsFAPbI₃ vs RbCsFAPbI₃ (Supplementary Fig. S71). To provide high efficiency PSCs, we optimized the recipe for perovskite fabrication for each condition, which is explained at experimental part. In terms of dispersed-solution SnO₂ (SnO₂_DS) vs. CBD SnO₂ (SnO₂_CBD), the initial efficiency of SnO₂_CBD with 2-phenylpyridine was 20.1% and remained at 18.7% after 936 hours of testing at 85 °C. In contrast, SnO₂_DS with 2-phenylpyridine showed an initial efficiency of 14.9% and retained 16.2% after 936 hours at 85 °C. In both cases, 2-phenylpyridine provided higher stability compared with 4-tert-butylpyridine. As for CsFAPbI₃ vs RbCsFAPbI₃, rubidium doped perovskite (RbCsFAPbI₃) achieved higher efficiency (24.2%) than CsFAPbI₃ (20.1%), however, stability with RbCsFAPbI₃ showed significantly dropped after 87 hours of 85 °C test. From this result, It is possible that rubidium doped perovskite itself has problems with thermal durability. Correa-Baena et al.

indicated that a dosage of 1% Rb was enough for causing segregation, which is a different kind of issue from those associated with additives¹⁰⁰. These results suggest that optimization of perovskite composition is also important to achieve high thermal stability.

3-5. Table 1 shows the code of different additives, which is not necessary to present it in main text.

Answer: Thank you for pointing out. We moved the code of different additives to supplementary information.

3-6. What is the author's numbering criterion for the additives? It seems rather chaotic in the article, as there is no specific order and some of the numbers are missing.

Answer: Thank you for your valuable comments. We've cleaned up the additive code to make it easier to read.

3-7. In fig.3, the SEM images of perovskite film based on different spiro-OMeTAD show different thickness, the authors should confirm this.

Answer: Thank you for your insightful comments. We examined the reproducibility of the Spiro-OMeTAD layer and confirmed that its thickness remained the same regardless of the additives. Variations in thickness can arise from the spin-coating process, which is influenced by the morphology of the underlying layers, such as SnO₂ and perovskite. By carefully preparing the SnO₂ and perovskite layers, we consistently obtained the same thickness for the Spiro-OMeTAD layer.

Figure 3. SEM images of perovskite solar cells

3-8. From the EQE curves and integrated Jsc in Fig.S69, the PCE of devices in Fig.5a-c should not obtain such high Jsc. The performance of these devices is overestimated.

Answer: Thank you for your suggestions. We have provided the IPCE data and the integrated current density of the highly efficient PCSs in Fig. 5c. The integrated current density was 25.14 mA/cm² for 4-tert-butylpyridine, 25.33 mA/cm² for 4-phenylpyridine, and 25.39 mA/cm² for 2-phenylpyridine. The difference between Jsc obtained from the IV curves and the integrated current density was within 4%, which is acceptable considering recent publications in *Nature Communications*, *Nature*, etc. These results have been included in the supplementary information.

Figure S72. IPCE of perovskite solar cells with 4-*tert*-butylpyridine (tBP), 4-phenylpyridine (A15), and 2-phenylpyridine (A82).

3-9. It is rarely that 9.8 mg/mL OAI was used for passivating the perovskite surface because it is hydrophobic to influence the deposition of Spiro-OMeTAD, the authors should confirm this.

Answer: Thank you for your valuable comments. OAI passivation has been introduced on some paper^[a-c], and we also confirmed that OAI passivation and its hydrophobic properties do not significantly influence the photovoltaic properties and stability of perovskite solar cells comparing to BAI passivation^[d] and PAI passivation^[e], which are less hydrophobic due to their shorter alkyl chains.

[a] Zhao, C., Zhou, Z., Almalki, M. *et al.* Stabilization of highly efficient perovskite solar cells with a tailored supramolecular interface. *Nat Commun* **15**, 7139 (2024).

[b] Kim, H., Yoo, S.M., Ding, B. *et al.* Shallow-level defect passivation by 6H perovskite polytype for highly efficient and stable perovskite solar cells. *Nat Commun* **15**, 5632 (2024).

[c] S. Zouhair *et al.* Employing 2D-Perovskite as an Electron Blocking Layer in Highly Efficient (18.5%) Perovskite Solar Cells with Printable Low Temperature Carbon Electrode. *Adv. Ener. Mater.*, **12**, 2200837, (2022)

[d] S. Mondal *et al.* Mixed 2D-cation passivation towards improved durability of perovskite solar cells and dynamics of 2D-perovskites under light irradiation and at high temperature, *Sustain. Ener. & Fuels*, **9**, 247, 2025.

[e] M. Aduhelaiqa *et al.* Mixed cation 2D perovskite: a novel approach for enhanced perovskite solar cell stability, *Sustain. Ener. & Fuels*, **6**, 2471 (2025).

Reviewer #4 (Remarks to the Author):

Answer: We thank the reviewers, including the Early Career Researcher who co-reviewed this manuscript, for their valuable comments and careful evaluation. Their suggestions have been very helpful in improving the clarity and quality of our work.

Response to reviewer

We appreciate so much for the effort and dedicated work of the editor and the reviewers. We are grateful for offering the opportunity to improve our work as major revision. We have judiciously taken all comments into account following the comments raised.

Comments on the reviewers: **in red color**

Our response: **in blue color**

Revised manuscript: **in yellow highlighted**

Reviewer #1 (Remarks to the Author):

The authors have added substantial new data that more convincingly support the earlier conclusions, and they have addressed the majority of my previous questions satisfactorily. The manuscript has improved considerably.

Answer: We sincerely appreciate your valuable comments. Your insightful suggestions have greatly enhanced the quality of our manuscript.

1-1. The attempt to determine the thin-film T_g by DSC was unsuccessful. Beyond DSC, there are several film-compatible methods to measure the glass transition temperature, such as variable-temperature spectroscopic ellipsometry (identifying the slope change in refractive index/thickness vs temperature) and dynamic mechanical analysis (DMA). Please consider adopting at least one of these approaches and report key parameters (heating rate, substrate, film thickness, etc.).

Answer: Thank you for your valuable comments. We are sorry but we do not have the equipment for variable-temperature spectroscopic ellipsometry or dynamic mechanical analysis (DMA). We spent a lot of time measuring T_g using DSC, but we found that measurements and analysis of mixed materials are quite difficult due to the complex interactions, which require too much time to make a conclusion. We would like to research T_g our next research topic.

Instead, to reveal the mechanism for thermal stability with novel additives, we investigated the effect of additives on elemental distribution using ToF-SIMS (Supplementary Fig. S80). The measured samples were prepared with the configuration FTO/SnO₂/Perovskite/Spiro-OMeTAD (with additives)/Au, following the same procedure as device fabrication. The Au layer was peeled off immediately before ToF-SIMS measurement. We compared 4-tert-butylpyridine and 2-phenylpyridine before and after thermal aging at 85 °C for 100 hours. Lithium migrated toward the ETL side and slightly accumulated at the surface, which is consistent with a previous report¹⁰³. The lithium distribution was not significantly affected by the choice of dopant, nor by the thermal test. From these results, 2-phenylpyridine does not appear to promote a more homogeneous lithium distribution.

For the fresh device with 4-tert-butylpyridine, the additive was mainly distributed in the Spiro-OMeTAD layer (Supplementary Fig. S80a). After the 85 °C test, however, 4-tert-butylpyridine had migrated substantially into the perovskite layer (Supplementary Fig. S80b). This migration is likely driven by its reaction with the perovskite, which is consistent with the by-product confirmed by XRD (Figure 4b). In contrast, 2-phenylpyridine was mainly localized in the Spiro-OMeTAD layer, and its distribution showed no significant change after the 85 °C test (Supplementary Fig. S80c,d). This stability may contribute to its beneficial effect on thermal stability, probably because 2-phenylpyridine does not react with the perovskite and therefore does not diffuse into the perovskite layer.

Furthermore, the ToF-SIMS results may help explain the absence of voids in the 2-

phenylpyridine-containing Spiro-OMeTAD layer after the 85 °C test. In the case of 4-tert-butylpyridine, its significant migration from Spiro-OMeTAD into the perovskite layer may have contributed to void formation within Spiro-OMeTAD (Supplementary Figure S81a,b). By contrast, the void-free morphology observed with 2-phenylpyridine can be attributed to the stability of its distribution (Supplementary Figure S81c,d).

Although it remains difficult to determine the precise mechanism of void formation from these results alone, they may provide valuable design guidelines for void-free HTMs.

The sentences were revised on page 23.

For further understanding of mechanism of the void-formation, we investigated the effect of additives on elemental distribution using ToF-SIMS. The measured samples were prepared with the configuration FTO/SnO₂/Perovskite/Spiro-OMeTAD (with additives)/Au, following the same procedure as device fabrication. The Au layer was peeled off immediately before ToF-SIMS measurement. We compared 4-tert-butylpyridine and 2-phenylpyridine before and after thermal aging at 85 °C for 100 hours. Lithium migrated toward the ETL side and slightly accumulated at the surface, which is consistent with a previous report¹⁰³. The lithium distribution was not significantly affected by the choice of dopant, nor by the thermal test. From these results, 2-phenylpyridine does not appear to promote a more homogeneous lithium distribution.

For the fresh device with 4-tert-butylpyridine, the additive was mainly distributed in the Spiro-OMeTAD layer. After the 85 °C test, however, 4-tert-butylpyridine had migrated substantially into the perovskite layer. This migration is likely driven by its reaction with the perovskite, which is consistent with the by-product confirmed by XRD (Figure 4b). In contrast, 2-phenylpyridine was mainly localized in the Spiro-OMeTAD layer, and its distribution showed no significant change after the 85 °C test. This stability may contribute to its beneficial effect on thermal stability, probably because 2-phenylpyridine does not react with the perovskite and therefore does not diffuse into the perovskite layer. These ToF-SIMS results may explain the absence of voids in the 2-phenylpyridine-containing Spiro-OMeTAD layer after the 85 °C test. In the case of 4-tert-butylpyridine, its significant migration from Spiro-OMeTAD into the perovskite layer may have contributed to void formation within Spiro-OMeTAD (Supplementary Figure S80a,b). By contrast, the void-free morphology observed with 2-phenylpyridine can be attributed to the

stability of its distribution (Supplementary Figure S80c,d).

Figure S80. ToF-SIMS depth profiles of (a) a fresh device with 4-tert-butylpyridine, (b) the same device after the 85 °C test, (c) a fresh device with 2-phenylpyridine, and (d) the same device after the 85 °C test.

Figure S81. Schematics images of void formation and void-free by additives. (a) a fresh device

with 4-tert-butylpyridine, (b) the same device after the 85 °C test, (c) a fresh device with 2-phenylpyridine, and (d) the same device after the 85 °C test.

1-2. On page 21, the manuscript states: “Interestingly, 3-phenylpyridine exhibited a slightly more compact structure than 4-phenylpyridine (A15) due to geometric distortion induced by its molecular orientation.” At present this “more compact/crowded” description is based on visual inspection, which is somewhat qualitative. I recommend quantifying the interaction/coordination strength (e.g., binding energy or dissociation barrier from calculations or experiments) to better substantiate the claimed tendency toward dissociation, and include methods and uncertainties in the main text.

Answer: Thank you for your pointing. To provide a more quantitative assessment rather than a qualitative description such as “more compact,” we calculated and compared the minimum edge lengths of the smallest boxes that could enclose each molecule. The minimum edge lengths were determined to be 10.93 Å for 4-tert-butylpyridine, 12.16 Å for 4-phenylpyridine, 8.01 Å for 3-phenylpyridine, and 9.06 Å for 2-phenylpyridine, respectively. Also, the desorption energy required to remove a coordinated molecule from lithium was calculated to be 10.93 Å for 4-tert-butylpyridine (tBP), 12.16 Å for 4-phenylpyridine (K27), 8.01 Å for 3-phenylpyridine (K28), and 9.06 Å for 2-phenylpyridine (K29), respectively, indicating that there is no relationship between thermal stability and ease of molecular desorption energy (page 22). These results led to a more quantitative evaluation of the manuscript.

The sentence was added on page 22.

The minimum edge lengths of the smallest boxes that could enclose each molecule were determined to be 10.93 Å for 4-tert-butylpyridine (tBP), 12.16 Å for 4-phenylpyridine (K27), 8.01 Å for 3-phenylpyridine (K28), and 9.06 Å for 2-phenylpyridine (K29), respectively.

1-3. The compound codes have been revised, but a few instances of the old codes remain. Please perform a thorough consistency check across the entire manuscript, including figure captions, tables, Supporting Information, and cross-references.

Answer: Thank you for your valuable comments. We have carefully checked the entire manuscript

to ensure accuracy.

Reviewer #2 (Remarks to the Author):

Recommendation: Accept

The authors have made detailed response to the reviewers' comments. The revision is greatly improved in the current form. I think this work is suitable for publication in NC.

Answer: Thank you very much for your valuable comments and for recommending acceptance. Your thoughtful feedback has significantly improved our manuscript.

Reviewer #3 (Remarks to the Author):

Recommendation: Accept

it can be accepted.

Answer: We sincerely appreciate your valuable comments and for recommending acceptance. Your insightful suggestions have greatly enhanced the quality of our manuscript.

Reviewer #4 (Remarks to the Author):

Answer: We thank the reviewers, including the Early Career Researcher who co-reviewed this manuscript, for their valuable comments and careful evaluation. Their suggestions have been very helpful in improving the clarity and quality of our work.